# Intramolecular domain dynamics regulate synaptic MAGUK protein interactions

**Nils Rademacher[1]\*, Benno Kuropka[2], Stella-Amrei Kunde[1], Markus C Wahl[3,4], Christian Freund[2], Sarah A Shoichet[1]\***

[1]Neuroscience Research Center, Charité-Universitätsmedizin Berlin, Berlin, Germany; [2]Institute of Chemistry and Biochemistry, Laboratory of Protein Biochemistry, Freie Universität Berlin, Berlin, Germany; [3]Institute of Chemistry and Biochemistry, Laboratory of Structural Biochemistry, Freie Universität Berlin, Berlin, Germany; [4]Macromolecular Crystallography, Helmholtz-Zentrum Berlin für Materialien und Energie, Berlin, Germany

**Abstract** PSD-95 MAGUK family scaffold proteins are multi-domain organisers of synaptic transmission that contain three PDZ domains followed by an SH3-GK domain tandem. This domain architecture allows coordinated assembly of protein complexes composed of neurotransmitter receptors, synaptic adhesion molecules and downstream signalling effectors. Here we show that binding of monomeric CRIPT-derived $PDZ_3$ ligands to the third PDZ domain of PSD-95 induces functional changes in the intramolecular SH3-GK domain assembly that influence subsequent homotypic and heterotypic complex formation. We identify PSD-95 interactors that differentially bind to the SH3-GK domain tandem depending on its conformational state. Among these interactors, we further establish the heterotrimeric G protein subunit Gnb5 as a PSD-95 complex partner at dendritic spines of rat hippocampal neurons. The PSD-95 GK domain binds to Gnb5, and this interaction is triggered by CRIPT-derived $PDZ_3$ ligands binding to the third PDZ domain of PSD-95, unraveling a hierarchical binding mechanism of PSD-95 complex formation.
DOI: https://doi.org/10.7554/eLife.41299.001

**\*For correspondence:**
nils.rademacher@charite.de (NR);
sarah.shoichet@charite.de (SAS)

**Competing interests:** The authors declare that no competing interests exist.

## Introduction

Excitatory synapses are the contact sites through which neurons communicate with each other. These synapses are asymmetric structures that are formed by pre- and postsynaptic terminals containing distinct sets of proteins. Incoming action potentials are converted into chemical signals (neurotransmitters) at presynaptic terminals, which subsequently pass through the synaptic cleft and are reconverted into electrical signals at postsynaptic sites (*Lisman et al., 2007*). These synaptic contacts are not static but are able to undergo structural changes and thereby modify neuronal network computation (*Nishiyama and Yasuda, 2015*). At postsynaptic sites, interacting proteins are densely packed into a sub-membrane structure called the postsynaptic density (PSD) (*Sheng and Hoogenraad, 2007*). Scaffold proteins of the PSD-95 family membrane-associated guanylate kinases (MAGUKs) are highly abundant components of the PSD and function as central regulators of postsynaptic organisation (*Zhu et al., 2016a*). PSD-95 family MAGUKs contain three PDZ domains that are known to directly interact with N-methyl-D-aspartate (NMDA) receptor C-termini (*Kornau et al., 1995*), α-amino-3-hydroxy-5-methyl-4-isoxazolepropionic acid (AMPA) receptor C-termini (*Leonard et al., 1998*), and AMPA receptor auxiliary subunit C-termini (*Dakoji et al., 2003*). The PDZ domains are followed by an SH3 - guanylate kinase (GK) domain tandem (*Funke et al., 2005*). The MAGUK SH3 domain lost its function to bind proline-rich peptides; instead it forms an intramolecular interaction with the GK domain (*McGee et al., 2001*). Similarly, the PSD-95 GK domain is atypical in that it is unable to phosphorylate GMP but has evolved as a protein interaction domain

(*Johnston et al., 2011*). Binding of known interactors to the GK domain typically involves residues of the canonical GMP-binding region (*Reese et al., 2007*; *Zhu et al., 2011*; *Zhu et al., 2016b*). This modular array of protein interaction domains allows PSD-95 MAGUKs to function as bidirectional organisers of synaptic function. First, neurotransmitter receptors can be incorporated or removed from postsynaptic membranes, depending on molecular interactions with these sub-membrane scaffold proteins. Second, together with other scaffold proteins at postsynaptic sites, they align downstream effectors and cytoskeletal proteins. Accordingly, PSD-95 family MAGUKs are essential for the establishment of long-term potentiation (LTP) by regulating the content of AMPA receptors at dendritic spines (*Ehrlich and Malinow, 2004*; *Opazo et al., 2012*; *Sheng et al., 2018*). In line with this is the observation that acute knockdown of PSD-95 MAGUKs leads to a decrease in postsynaptic AMPA and NMDA receptor-mediated synaptic transmission as well as a reduction in PSD size (*Chen et al., 2015*). Taken together, exploring protein complex formation directed by PSD-95 MAGUK family members is of central importance for understanding regulation of synaptic transmission. We have previously shown that the synaptic MAGUK protein PSD-95 oligomerises upon binding of monomeric CRIPT-derived $PDZ_3$ ligands (ligands that specifically bind to the third PDZ domain) (*Rademacher et al., 2013*) and speculated that ligand - $PDZ_3$ domain binding induces conformational changes in the C-terminal domains that lead to complex formation. Our initial observations of PDZ ligand-induced effects in PSD-95 MAGUK proteins have recently been supported by other studies (*Zeng et al., 2016*; *Zeng et al., 2018*).

In this study, we use a bimolecular fluorescence complementation (BiFC) assay to show that PSD-95 oligomerisation can be triggered by Neuroligin-1 (NLGN1) and that this is dependent on the C-terminal SH3-GK domain tandem. Moreover, we identify new interaction partners of PSD-95 C-terminal domains by quantitative mass spectrometry. We provide evidence that the heterotrimeric G protein subunit Gnb5 is a novel GK domain interactor and that its ability to bind to PSD-95 is likewise promoted by binding of a ligand to the PSD-95 $PDZ_3$ domain.

## Results

### Neuroligin-1 binding to PSD-95 $PDZ_3$ domains facilitates oligomerisation guided by the PSG module

We are interested in the functional coupling of $PDZ_3$ domains with the adjacent SH3-GK domain tandem in the synaptic scaffold protein PSD-95 (termed PSG module, see *Figure 1A* for domain structure) and the relevance of ligand - $PDZ_3$ domain interactions for PSD-95 complex formation. To explore this idea, we built on our previous work with tagged cytosolic CRIPT-derived $PDZ_3$ ligands (*Rademacher et al., 2013*) and we have now designed a cell-based assay to directly monitor the proximity of PSD-95 molecules by bimolecular fluorescence complementation (BiFC). Expression constructs of PSD-95 were fused to non-fluorescent halves of EYFP (N-terminal half = YN and C-terminal half = YC) and coexpressed with the established PDZ domain ligand Neuroligin-1 (NLGN1) in HEK-293T cells. NLGN1 is a synaptic adhesion molecule that specifically binds to the third PDZ domain of PSD-95 (*Irie et al., 1997*). Coexpression of the *per se* non-fluorescent PSD-95-YN and PSD-95-YC constructs (together referred to as WT/WTsplitEYFP) with full-length NLGN1 led to the formation of multimolecular fluorescent PSD-95 complexes that were located at the cell membrane, recapitulating the natural localisation of the endogenous protein complexes (*Figure 1B*), and highlighting that the PSD-95 C-termini (which harbour the splitEYFP tags) are in close proximity to each other in these complexes.

In order to quantify the formation of these fluorescent protein complexes, we used flow cytometry. In initial experiments (*Figure 1C*, see also *Figure 1—figure supplement 1* for gating strategy), we validated that efficient refolding of splitEYFP halves fused to wild-type PSD-95 indeed relied on the presence of a PDZ-binding ligand that interacts with PSD-95. For comparison, we took advantage of the synaptic cell adhesion molecule SynCAM1. SynCAMs, like NLGNs, are trans-synaptic membrane proteins; however, they differ from NLGN family members at the C-terminus and bind exclusively to class II PDZ domains (*e.g.* CASK, MPP2) rather than class I PDZ domains (such as PSD-95 PDZ domains) (*Biederer et al., 2002*). When we express the splitEYFP-tagged wild-type PSD-95 molecules together with SynCAM1, we do not induce refolding of splitEYFP (*Figure 1C*), suggesting that it is indeed the PDZ domain interactions between the transmembrane protein and wild-type

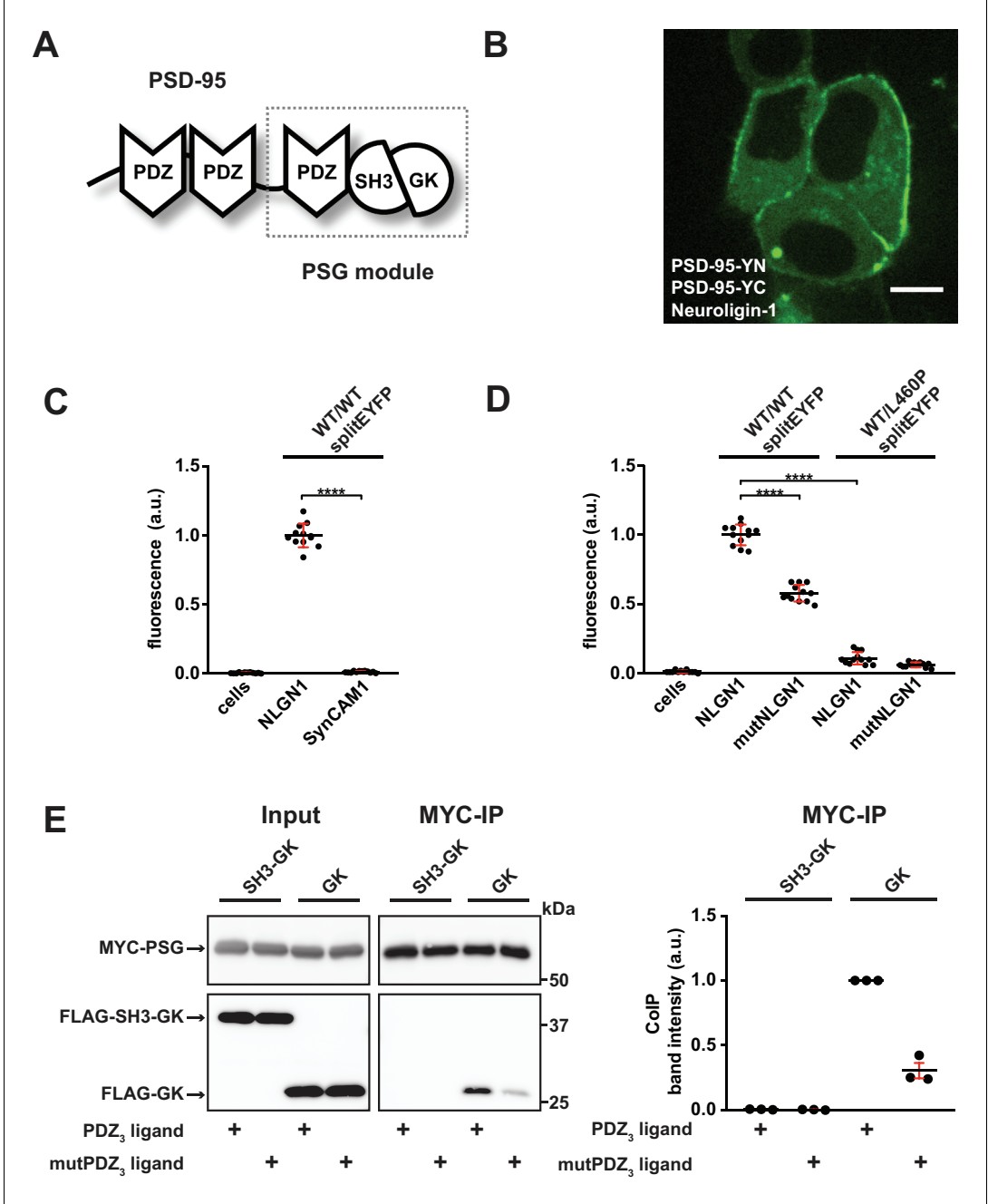

**Figure 1.** PDZ$_3$ ligand-induced dynamics in the PDZ$_3$-SH3-GK module facilitate oligomerisation. (**A**) Schematic representation of the PSD-95 domain organisation. PSD-95 contains three PDZ domains followed by a SH3-GK domain tandem. The PSG module (PDZ$_3$-SH3-GK) is common to the MAGUK protein family. (**B**) Live-cell microscopy of HEK-293T cells transfected with PSD-95-YN, PSD-95-YC and full-length Neuroligin-1 reveals a membrane associated localisation of the refolded complex (transfection corresponding to WT/WTsplitEYFP plus NLGN1 in *Figure 1C,D*). Scale bar: 10 μm. (**C**, **D**) PSD-95 oligomerisation assay based on BiFC. HEK-293T cells were triple-transfected with the displayed DNA constructs and EYFP refolding was assessed by flow cytometry. Formation of oligomeric fluorescent complexes is effective in the presence of wild-type Neuroligin-1 (NLGN1). (**C**) Fluorescence is almost not detectable by coexpression of SynCAM1 (SynCAM1 is not binding to PSD-95 PDZ domains) (**D**) Fluorescence is reduced by either site-directed mutagenesis of the NLGN1 PDZ$_3$ ligand C- terminus (mutNLGN1: TTRV ▶ TARA), or a targeted amino acid exchange in the PSD-95 SH3 domain (L460P). (**C**, **D**) The dot plots indicate mean values (black horizontal bar) with SD (red vertical bar), based on twelve individual measurements (dots) that originate from four independent experiments (results from each experiment are triplicates for each DNA construct combination). Data were analysed by one-way ANOVA/Sidak's multiple comparisons test. ****p<0.0001. (**E**) MYC-PSG and FLAG-SH3-GK or FLAG-GK were coexpressed together with either CRIPT-derived PDZ$_3$ or mutPDZ$_3$ ligand constructs. Upon MYC-PSG IP, proteins were analysed by western blot with αFLAG antibodies. Coexpression of the CRIPT-derived PDZ$_3$ ligand enhanced the coIP of PSG and GK, whereas coIP of PSG and SH3-GK was

*Figure 1 continued on next page*

*Figure 1 continued*

negligible regardless of whether or not the CRIPT-derived PDZ$_3$ ligand construct was coexpressed. The western blot shown (left side) is a representative example of three independent experiments; the corresponding quantification of coIP band intensities from these three experiments is shown in the dot plot on the right side indicating mean values ± SEM.

DOI: https://doi.org/10.7554/eLife.41299.002

The following source data and figure supplements are available for figure 1:

**Source data 1.** Source data for *Figure 1C,D*.
DOI: https://doi.org/10.7554/eLife.41299.003
**Source data 2.** Source data for *Figure 1E*.
DOI: https://doi.org/10.7554/eLife.41299.004
**Figure supplement 1.** FACS plots for *Figure 1C,D*.
DOI: https://doi.org/10.7554/eLife.41299.005
**Figure supplement 2.** Supplement for *Figure 1D*.
DOI: https://doi.org/10.7554/eLife.41299.006

PSD-95 that facilitate the observed formation of multimolecular PSD-95 complexes in which the splitEYFP halves refold efficiently.

We validated this idea in a second set of experiments by taking advantage of a targeted mutation of the PDZ-binding sequence of NLGN1: we generated mutant NLGN1 variants that carry two alanine substitutions within the C-terminal PDZ$_3$ ligand sequence (mutNLGN1: C-terminus TTRV ▶ T**A**R**A**). Upon coexpression of this mutNLGN1 with splitEYFP-tagged PSD-95 molecules, the detected fluorescence intensity decreased by approximately 40% relative to that after coexpression with wild-type NLGN1 (*Figure 1D*, see also *Figure 1—figure supplement 1* for gating strategy), supporting the idea that indeed the ligand - PDZ domain interaction is critical for PSD-95 complex formation in our assay. Finally, we investigated the hypothesis that the natural PSD-95 SH3-GK conformation is also important in this PDZ ligand-mediated scaffolding process. Leucine 460 is an internal SH3 domain residue and the L460P mutation has been shown to specifically disrupt the intramolecular SH3-GK domain interaction (*Figure 1—figure supplement 2*) (*McGee and Bredt, 1999*; *Shin et al., 2000*) that is one of the hallmark features of MAGUK proteins (*Tavares et al., 2001*). Interestingly, this amino acid exchange does not interfere with PDZ$_3$ ligand binding (*Rademacher et al., 2013*) but strongly abolishes PSD-95 complex assembly, as observed by EYFP refolding (*Figure 1D*). We assume that in the context of the full-length protein, the L460P mutation likewise weakens the (intramolecular) interaction between the SH3 and GK domains, which would then result in a constitutively 'open' conformation. The profound negative effect that we observe following a targeted amino acid exchange in the SH3 domain highlights the importance of the SH3-GK domain tandem for its involvement in regulated PSD-95 oligomerisation.

## Binding of a CRIPT-derived PDZ ligand to PSD-95 PDZ$_3$ facilitates an 'open' SH3-GK state that frees both domains for binding in *trans*

In line with our BiFC assay results, we have previously reported that PSD-95 constructs (full-length and the isolated PSG module) efficiently oligomerise and coprecipitate upon binding of a PDZ$_3$ ligand (*Rademacher et al., 2013*). Moreover, the observation by NMR spectroscopy that the PSG module forms a dynamic modular entity (*Zhang et al., 2013*) led us to hypothesise that this type of ligand binding to PDZ$_3$ might influence intramolecular SH3-GK domain assembly, facilitating the formation of domain swapped oligomers (*McGee et al., 2001*; *Ye et al., 2018*). For these experiments, we again took advantage of the PSD-95 interactor CRIPT, which is a cytosolic protein that has been shown to interact predominantly with the third PDZ domain of PSD-95 (*Niethammer et al., 1998*). Specifically, we asked whether the ligand - PDZ$_3$ domain interaction might release the intramolecular SH3-GK domain assembly, thereby allowing other domains and proteins to interact in *trans*. To explore this idea, we assessed which PSD-95 domains are able to interact in *trans* upon binding of a CRIPT-derived PDZ$_3$ ligand to proteins that harbour the PSG module, using a coimmunoprecipitation experiment designed accordingly.

Our strategy involves expression of the PSD-95 PSG module together with mCherry-tagged PDZ$_3$ ligands, consisting of the last 10 amino acids of CRIPT (DTKNYKQTSV) (*Niethammer et al., 1998*). As a control, comparable constructs carrying two amino acid exchanges within the PDZ$_3$ ligand

sequence (DTKNYKQ**A**S**A**) were used. In our hands, this mutation in the CRIPT C-terminus almost completely abrogates binding to $PDZ_3$, as observed by coIP (*Rademacher et al., 2013*), making it an ideal control for our coIP approach that targets the ligand - $PDZ_3$ interaction. Upon triple trans-fection with either a GK or an SH3-GK domain construct and the CRIPT-derived $PDZ_3$ ligand, the PSG modules were precipitated, and copurified proteins were analysed by western blot (*Figure 1E*). The SH3-GK construct did not coprecipitate with the PSG module regardless of whether it was coex-pressed with wild-type or mutant CRIPT-derived $PDZ_3$ ligands. This may be due to a constitutive intramolecular association of the SH3 and GK domains, leading to a 'closed' SH3-GK assembly, with no ability to bind a PSG module in *trans*. The GK domain alone, however, coprecipitated effectively with the PSG module, when expressed in the presence of functional CRIPT-derived $PDZ_3$ ligands. These data suggest that binding of a CRIPT-derived $PDZ_3$ ligand renders the PSG module 'interac-tion-competent'; that is it facilitates formation of a conformational state in which it is able to bind isolated GK domain constructs in *trans*. In this experiment, the intramolecular SH3-GK domain assembly resembles the 'interaction-incompetent' state, and the SH3 domain autoinhibits the GK domain's interaction activity.

## The SH3-GK assembly state influences PSD-95 interactions

Based on the above results, we propose that PSD-95 C-termini can adopt different functional states depending on whether or not CRIPT-derived $PDZ_3$ ligands are bound to PSD-95 $PDZ_3$ domains, *i.e.* ligand binding induces a loosening of the intramolecular SH3-GK domain assembly and renders the SH3-GK domain tandem 'interaction-competent'. In order to identify interactors that differentially bind to PSD-95 C-termini in an 'open state' versus PSD-95 molecules where the GK domain is autoin-hibited by an intramolecular interaction with the adjacent SH3 domain, we utilised a quantitative proteomics strategy. In a reductionist approach, we mimic the open and closed states with different bacterially expressed GST fusion proteins: a GST-GK construct serves as the 'interaction-competent' GK domain state, whereas a GST-SH3-GK domain fusion protein reflects the autoinhibited domain assembly. By performing GST pull-downs from crude synaptosome preparations followed by quanti-tative mass spectrometric analysis we aimed to identify novel proteins that preferentially bind to the 'open' or 'closed' state of the PSD-95 C-terminal domains (*Figure 2A*).

Bacterially expressed GST-GK vs. GST-SH3-GK constructs were incubated with solubilised pro-teins from crude synaptosome preparations of whole rat brains in triplicates. Interacting proteins were eluted from the beads and separated by SDS-PAGE. Enzymatic $^{16}O/^{18}O$-labelling during tryp-tic in-gel digestion was used for relative quantification of proteins by nanoLC-MS/MS analysis. Pro-teins enriched by GST-GK were labelled light ($^{16}O$), while proteins enriched by GST-SH3-GK were labelled heavy ($^{18}O$). In total, we reproducibly identified and quantified 278 proteins (*Figure 2— source data 1*). Remarkably, 230 ($\approx$ 82%) of these have been reported to be present in mouse and/ or human cortical postsynaptic density fractions (*Bayés et al., 2012*). Moreover, we also identified the known GK-domain interacting proteins Map1A (*Reese et al., 2007*), Mark2 (*Wu et al., 2012*), Dlgap2 (*Takeuchi et al., 1997*) and Srcin1/p140CAP (*Fossati et al., 2015*), validating the general success of our approach. Potential binders to the GST-GK construct are expected to be enriched in their light form (L/H ratio >1), while binders to the GST-SH3-GK construct are expected to be enriched in their heavy form (L/H ratio <1). Unexpectedly, we isolated several heterotrimeric G pro-tein subunits enriched in the protein fractions that bind preferentially to the GST-GK construct (*Figure 2B*). Of special interest was the guanine nucleotide binding protein beta 5 (Gnb5), which is a signalling effector downstream of GPCRs that exhibits inhibitory activity in neurons (*Xie et al., 2010*; *Ostrovskaya et al., 2014*). Gnb5 contains an N-terminal α-helix followed by a β-sheet propeller composed of seven WD-40 repeats (*Cheever et al., 2008*). Gnb5 is specifically expressed in brain (*Watson et al., 1994*) and mutations in the Gnb5 gene cause a multisystem syndrome with intellec-tual disability in patients (*Lodder et al., 2016*).

## Gnb5 is a novel synaptic PSD-95 complex partner

In order to verify Gnb5 as a potential binding partner from the above mass spectrometry result, we performed a GST pull-down from crude rat brain synaptosomes and analysed the associated pro-teins by western blot (*Figure 3A*). We could not detect Gnb5 in the bead control pull-down lane and almost no Gnb5 was detectable in the GST-SH3-GK lane. However, a clear Gnb5 signal was

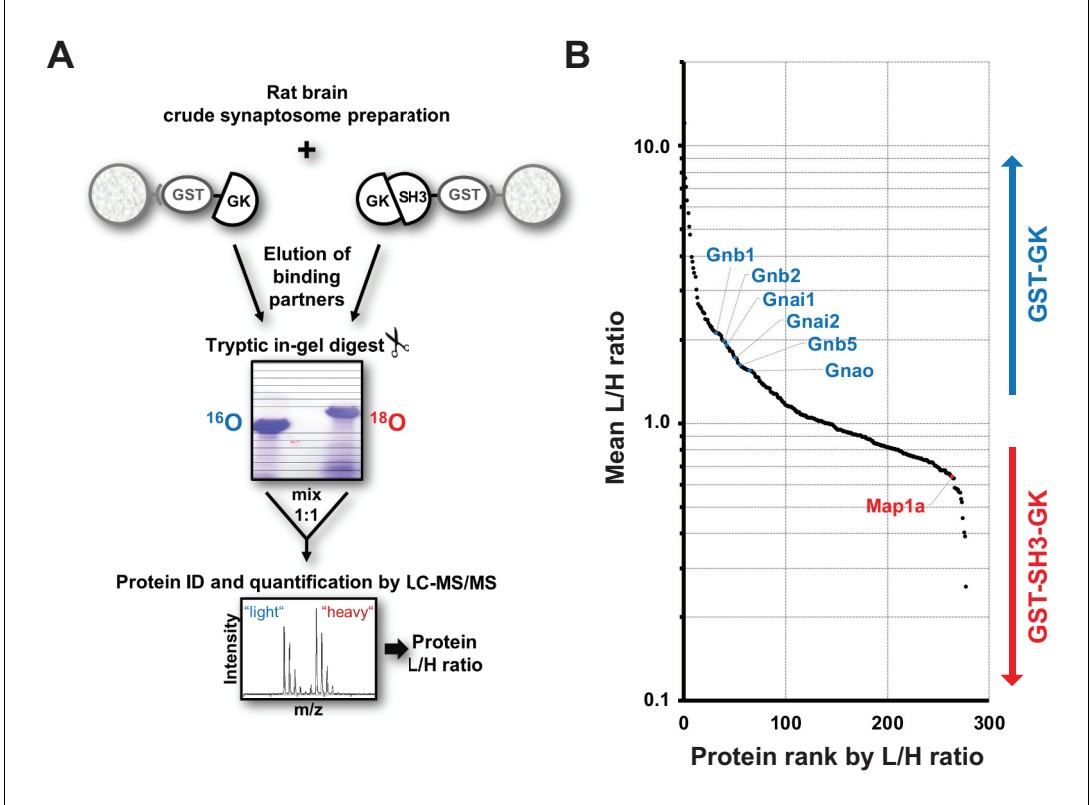

**Figure 2.** Identification of interactors that differentially bind to PSD-95 C-terminal domains. (**A**) Schematic representation of the quantitative mass spectrometry experiment to identify PSD-95 GK domain interactors from crude rat synaptosomes by GST pull-down of bacterially expressed GST-GK or GST-SH3-GK constructs and $^{18}$O-labeling. (**B**) GST pull-downs were performed in triplicates and 278 interacting proteins that passed our threshold settings were identified and quantified by mass spectrometry. Proteins are ranked by their mean L/H ratio indicating preferential enrichment with either GST-GK or GST-SH3-GK constructs. The heterotrimeric G protein subunit Gnb5 was found to be enriched in the GST-GK fraction relative to the GST-SH3-GK fraction and selected for further studies.

DOI: https://doi.org/10.7554/eLife.41299.007

The following source data is available for figure 2:

**Source data 1.** Source data for *Figure 2B*.
DOI: https://doi.org/10.7554/eLife.41299.008

present in the GST-GK lane, supporting our quantitative mass spectrometry results and suggesting that a Gnb5 - GK domain interaction is favoured over a Gnb5 - SH3-GK domain interaction. Additionally, we observed a preferred interaction of overexpressed Gnb5 with the isolated GK domain compared to SH3-GK domain constructs in COS-7 cells. Upon IP of Gnb5 tagged with the green fluorescent protein Clover (*Lam et al., 2012*) using αGFP antibodies, the isolated GK domain coprecipitates far more efficiently than does the SH3-GK domain (*Figure 3B*).

Our *in vitro* experiments clearly indicate that Gnb5 is an interactor of PSD-95 C-terminal domains. However, our interaction data do not clearly indicate in which subcellular compartment Gnb5 and the Gnb5 - PSD-95 complex is located. To explore this, we immunostained cultures of dissociated cells from rat hippocampi and analysed the subcellular distribution of endogenous proteins. We stained fixed cultures (DIV21) with antibodies against Gnb5 and costained for the dendritic marker MAP2 and for PSD-95. Gnb5 staining was present in neuronal dendrites, where the signal overlaps with the PSD-95 staining (*Figure 3C*). Additionally, we stained neurons with antibodies against Gnb5, MAP2 and the presynaptic marker Synapsin. In these experiments, the Gnb5 signal is adjacent to the presynaptic Synapsin signal (*Figure 3C*). Together, these findings strongly support the idea that Gnb5 and PSD-95 are protein complex partners at postsynaptic sites of hippocampal neurons.

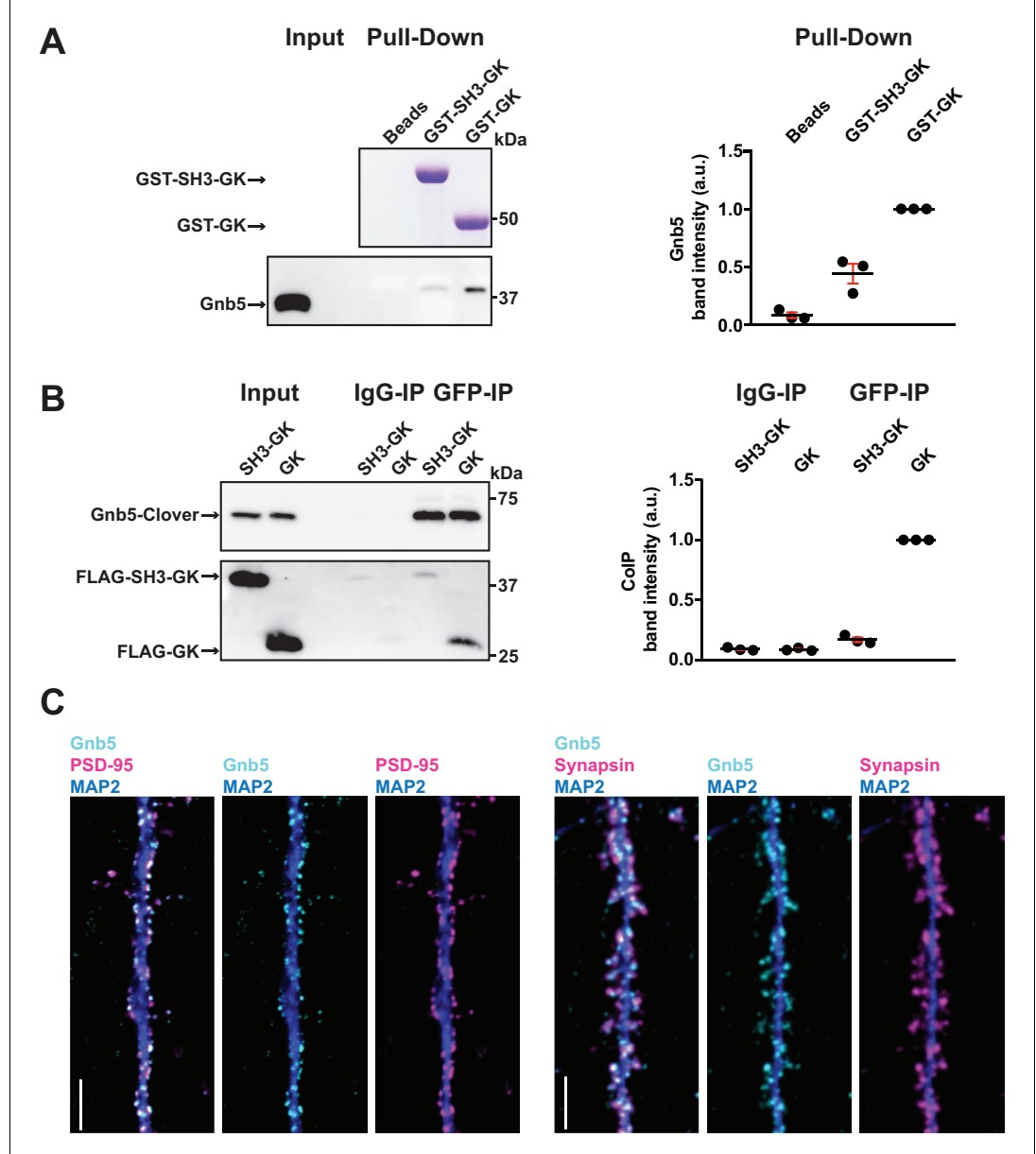

**Figure 3.** The heterotrimeric G protein subunit Gnb5 is a novel PSD-95 interactor. (**A**) GST pull-down from crude synaptosomal proteins (comparable amounts of GST tagged proteins observable by Coomassie, upper panel) enabled comparison of Gnb5 binding to the GK domain alone versus the SH3-GK domain. Gnb5 is effectively enriched in the GST-GK pull-down compared to bead controls or GST-SH3-GK pull-downs, as observed by western blot with a commercially available αGnb5 antibody (lower panel). The GST pull-down shown on the left side is a representative example of three independent experiments; the corresponding quantification of copurified Gnb5 band intensities from these three experiments is shown in the dot plot on the right side indicating mean values ± SEM. (**B**) CoIP experiment of tagged Gnb5 (Gnb5-Clover) with tagged SH3-GK or GK (FLAG-SH3-GK or FLAG-GK). Immunoprecipitation of Gnb5-Clover with αGFP antibody efficiently copurified the GK-domain construct (observed via western blot with αFLAG antibodies, lower panel). The western blot shown (left side) is a representative example of three independent experiments; the corresponding quantification of coIP band intensities from these three experiments is shown in the dot plot on the right side indicating mean values ± SEM. (**C**) Cultures of rat hippocampal neurons (E18) were fixed at DIV21 and stained for Gnb5 together with the dendritic marker MAP2 (microtubule-associated protein 2) and either the postsynaptic protein PSD-95 (left panel) or the presynaptic marker Synapsin (right panel) and respective fluorescent secondary antibodies, and visualised by confocal microscopy. Scale bars: 5 µm.

DOI: https://doi.org/10.7554/eLife.41299.009

The following source data is available for figure 3:

**Source data 1.** Source data for *Figure 3A,B*.

DOI: https://doi.org/10.7554/eLife.41299.010

## Regulation of PSD-95 complex formation

Our data indicate that Gnb5 interacts differentially with PSD-95 C-terminal constructs and we observe that PSD-95 and Gnb5 exhibit overlapping expression at postsynaptic sites. We next set out to determine if the PSD-95 - Gnb5 interaction is indeed influenced by the presence of synaptic PDZ$_3$ ligands, as we initially hypothesised. We coexpressed PSD-95 with CRIPT-derived PDZ$_3$ ligand constructs as in previous experiments, together with Gnb5. Following IP of PSD-95, the precipitates were analysed by western blot: the presence of these PDZ$_3$ ligands indeed triggered coimmunoprecipitation of Gnb5 and PSD-95, which supports the idea that ligand binding to PDZ$_3$ indirectly affects protein-protein interactions at neighbouring domains. Gnb5 lacking the N-terminal α-helix (shortGnb5) coprecipitated somewhat less efficiently than the full-length protein (*Figure 4A*), suggesting that this N-terminal region of Gnb5 (amino acids 1–33) is important for the PDZ$_3$ ligand-mediated interaction with PSD-95.

Next, we asked if the PSD-95 PSG module is sufficient to bind to Gnb5 in a ligand-triggered mode. We coexpressed a PSG expression construct together with Gnb5 and CRIPT-derived PDZ$_3$ ligand constructs (wild-type or mutant) and performed pull-downs of the PSG constructs or unspecific IgGs as a control. Upon analysis of the precipitates by western blot, we detected a robust coIP of Gnb5 with the PSG module construct in the presence of this type of PDZ$_3$ ligand (*Figure 4B*). Clearly, the PSG module is sufficient for ligand-triggered coimmunoprecipitation of Gnb5.

Our comparative mass spectrometry results for Gnb5, together with subsequent PSD-95 coimmunoprecipitation data, support the idea that ligand binding can influence the PSD-95 PSG module such that its protein interaction profile resembles that of the isolated GK domain, that is it differs from the SH3-GK domain tandem with regard to protein–protein interactions (see *Figure 1E*). In summary, we propose that binding of a PDZ$_3$ ligand weakens the intramolecular SH3-GK domain association, which then enables the individual SH3 and GK domains to participate in *trans* interactions with other molecules. To test this model, we took advantage of the PSD-95 L460P mutation, which is known to disrupt the well-characterised intramolecular SH3-GK domain assembly, thus aberrantly releasing the GK domain from its SH3 domain-mediated inhibition. Upon coexpression of wild-type or mutant (L460P) PSG proteins together with Gnb5, we performed pull-downs of the PSG proteins and comparatively assessed coprecipitation of Gnb5. Gnb5 did not coprecipitate efficiently with the wild-type PSG module but was effectively coprecipitated by the PSG module harbouring the L460P mutation that disrupts the intramolecular SH3-GK domain interaction (*Figure 4C*). We conclude that Gnb5 is interacting with the PSD-95 PSG module in one of two possible modes. Gnb5 could bind at GK domain sites that are directly occupied by the neighbouring SH3 domain (and thereby compete with the SH3 domain for interaction with the GK domain). Alternatively, Gnb5 could bind to GK domain sites on distant surfaces (e.g. the canonical GMP-binding region) that are not directly influenced by intramolecular SH3-GK interactions but might be allosterically regulated by changes to the PSG module.

## GK domain interactions are differentially regulated

In order to explore these two possibilities in more depth, we took advantage of established knowledge on the structure of GK domains and information on previously identified GK-interacting proteins. The GK domain of PSD-95 has evolved from an enzyme that catalyses the phosphorylation of GMP to an enzymatically inactive protein interaction domain. Interestingly, various PSD-95 GK-interacting proteins bind to the canonical GMP-binding region, and by exchanging arginine 568 (which is situated in the ancestral GMP-binding site) to alanine (R568A), these interactions can be specifically disrupted (*Reese et al., 2007*). In order to gain insight into the nature of the binding of Gnb5 to the PSD-95 GK domain, we compared PSD-95 - Gnb5 binding to PSD-95 - GKAP binding. GKAP ('GK'-associated protein, also referred to as SAPAP1 or DLGAP1) is an established synaptic GK domain binder (*Kim et al., 1997*) whose interaction involves the GMP-binding region (*Zhu et al., 2017*). These ideas are also validated by our own coimmunoprecipitation experiments: GKAP can be efficiently coprecipitated upon pull-down of either the isolated GK domain or an intact PSG module, whereas a recombinant PSG module harbouring the GMP binding site mutation R568A fails to precipitate GKAP (*Figure 5A*). In experiments with PSD-95 and Gnb5, however, the same mutation had no effect on coprecipitation of Gnb5 (*Figure 5B*), suggesting that GKAP and Gnb5 proteins bind to PSD-95 GK domains in fundamentally different ways.

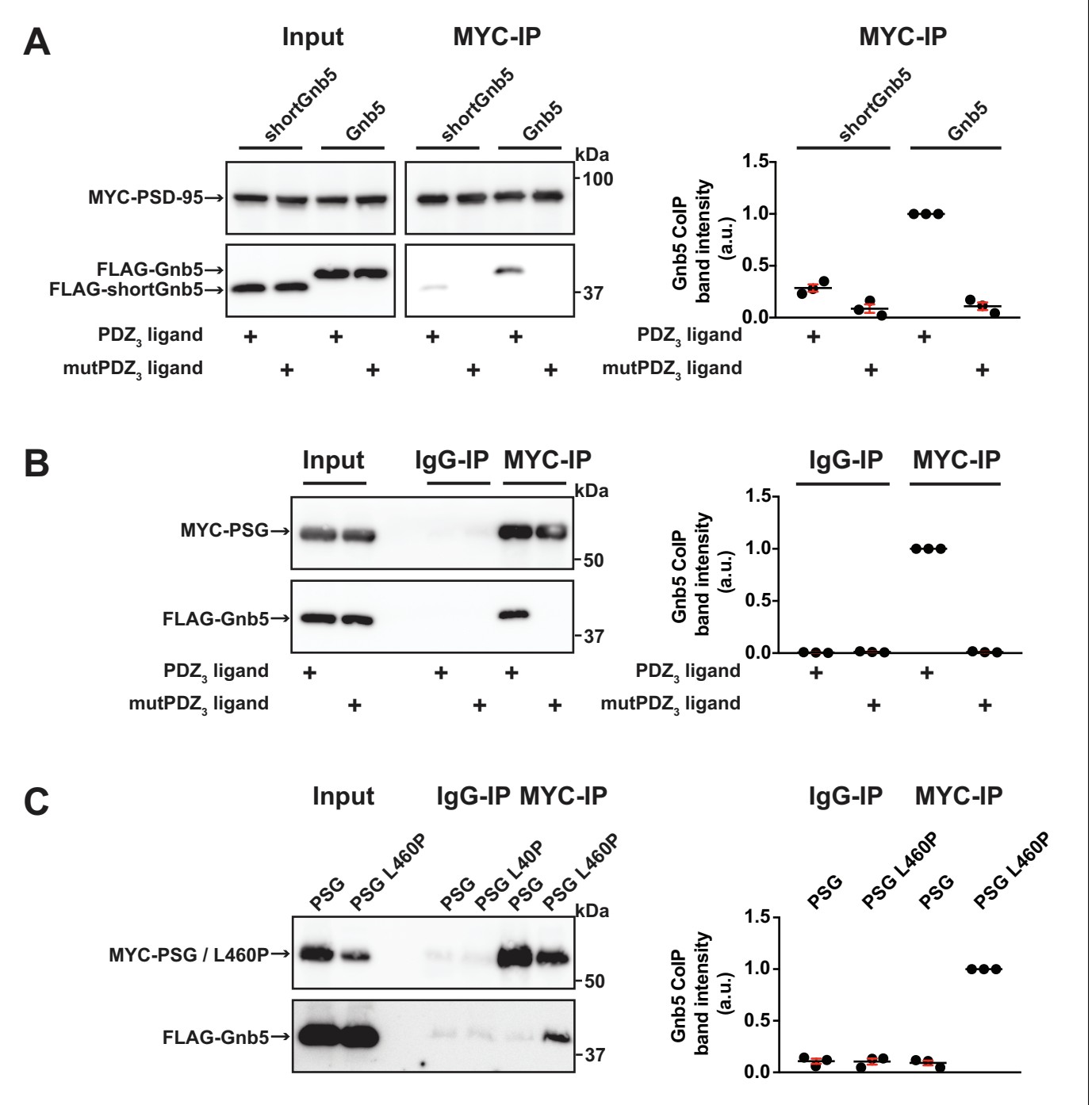

**Figure 4.** Gnb5 - PSD-95 complex formation is regulated by CRIPT-derived PDZ$_3$ ligand binding. For this figure, western blots shown on the left side are representative examples of three independent experiments; the corresponding quantification of coIP band intensities from these three experiments are shown in the dot plots on the right side indicating mean values ± SEM. (**A**) MYC-PSD-95 and FLAG-Gnb5 or FLAG-shortGnb5 were coexpressed with either CRIPT-derived PDZ$_3$ or mutPDZ$_3$ ligand constructs. MYC-PSD-95 was precipitated and proteins were analysed by western blot with αFLAG antibodies. Coexpression of the CRIPT-derived PDZ$_3$ ligand facilitated the coIP of PSD-95 and Gnb5, coIP with the shortGnb5 construct (N-terminal truncation) was much less efficient. In the presence of the CRIPT-derived mutPDZ$_3$ ligand, coprecipitated proteins were not detectable. (**B**) CoIP of MYC-PSG and FLAG-Gnb5 together with either CRIPT-derived PDZ$_3$ or mutPDZ$_3$ ligand constructs. The presence of CRIPT-derived PDZ$_3$ ligand constructs facilitated coprecipitation of PSG and Gnb5 (see comparative western blot with αFLAG antibodies, lower panel). (**C**) Coexpression of MYC-PSG or MYC-PSG L460P with FLAG-Gnb5 and subsequent MYC IP. PSG L460P IP efficiently copurifies Gnb5 (observed by western blot with αFLAG antibodies).

DOI: https://doi.org/10.7554/eLife.41299.011

*Figure 4 continued on next page*

*Figure 4 continued*

The following source data is available for figure 4:

**Source data 1.** Source data for *Figure 4*.
DOI: https://doi.org/10.7554/eLife.41299.012

We next tested whether the GKAP - PSD-95 association could be influenced by CRIPT-derived PDZ$_3$ ligands that bind to PSD-95, as we observed previously for Gnb5 (see *Figures 4A, B* and *5B*). The presence of PDZ$_3$ ligands did not substantially influence the GKAP interaction: PSD-95 binds GKAP regardless of whether wild-type or mutant CRIPT-derived PDZ$_3$ ligands were present (*Figure 5C*). These data provide further evidence that the GKAP - GK domain binding mode differs substantially from the Gnb5 - GK interaction mode.

Importantly, our data support a model in which ligand binding to PDZ$_3$ results in a conformational change of the 'resting' intramolecular SH3-GK interaction that is common to MAGUK proteins. This conformational alteration is reflected by a change in the availability of specific GK surfaces for protein-protein interactions (*Figure 6A*). In the resting state, the external GK surface harbouring the classical GMP binding site is available for protein-protein interaction such as those with the well-known PSD-95 interactors GKAP and MAP1a. However, upon ligand binding, other GK surfaces become accessible for protein-protein interactions. A subset of synaptic GK-interacting proteins – in particular Gnb5, and perhaps other proteins enriched in our pool of interacting proteins that bind preferentially to GK rather than to SH3-GK – bind to these surfaces of the GK domain (*Figure 6B*).

## Discussion

The molecular basis for the dynamic regulation of synaptic transmission is dependent on the assembly and disassembly of protein complexes (*Yokoi et al., 2012*; *Lautz et al., 2018*). It is also well established that activity-dependent changes in synaptic protein networks depend on phosphorylation (*Opazo et al., 2010*; *Araki et al., 2015*; *Li et al., 2016*) and other post-translational modifications, such as palmitoylation (*El-Husseini et al., 2002*; *Fukata et al., 2013*). Recently, it has been reported that the minimal requirement for the amplification of synaptic signals is the interaction of glutamate receptor auxiliary subunits with postsynaptic scaffold proteins; specifically the interaction between different PDZ ligand C-termini and synaptic MAGUKs can trigger a common molecular mechanism necessary for the induction of long-term potentiation downstream of glutamate receptors (*Sheng et al., 2018*).

In this study, we focussed on the postsynaptic scaffold protein PSD-95, which plays a central role in activity-dependent synapse regulation (*Ehrlich et al., 2007*). It is established that protein complex formation guided by PSD-95 PDZ and GK domains can be reversibly regulated by phosphorylation (*Sumioka et al., 2010*; *Zhu et al., 2017*), and at postsynaptic membranes, various PDZ ligand C-termini of multimeric receptor complexes are available to form multivalent interactions with scaffold proteins (*Schwenk et al., 2012*). In previous work, we showed that binding of CRIPT-derived ligands to PDZ$_3$ of PSD-95 promoted formation of PSD-95 multimers (*Rademacher et al., 2013*). More recently, it was shown by others that binding of a SynGAP-derived PDZ ligand peptide was sufficient to induce PSD-95 PSG construct dimerisation (*Zeng et al., 2016*), and the same group subsequently explored how SynGAP-induced conformational coupling between the PDZ$_3$ domain and the SH3-GK module plays a role in this process (*Zeng et al., 2018*).

Here we identify synaptic interactors whose association with PSD-95 is influenced by the conformational state of the PSD-95 C-terminus. Among these proteins, we focussed further on Gnb5. Gnb5 acts in a heterodimeric complex together with RGS7 to functionally couple GIRK channels to GABA$_B$ receptors (*Xie et al., 2010*; *Ostrovskaya et al., 2014*). Upon activation of GABA$_B$ receptors, the Gnb5-RGS7 complex promotes fast deactivation of GIRK channels (*Fajardo-Serrano et al., 2013*). Interestingly, the interaction between RGS proteins and Gnb5 involves the N-terminal helical domain of Gnb5 and the G protein gamma-subunit-like (GGL) domain of RGS proteins (*Cheever et al., 2008*). In our experiments, the protein complex formation between PSD-95 and Gnb5 likewise involved the N-terminal helix of Gnb5: its deletion (see shortGnb5) strongly inhibited the interaction with PSD-95. It will be interesting to explore if PSD-95 competes with RGS7 for Gnb5

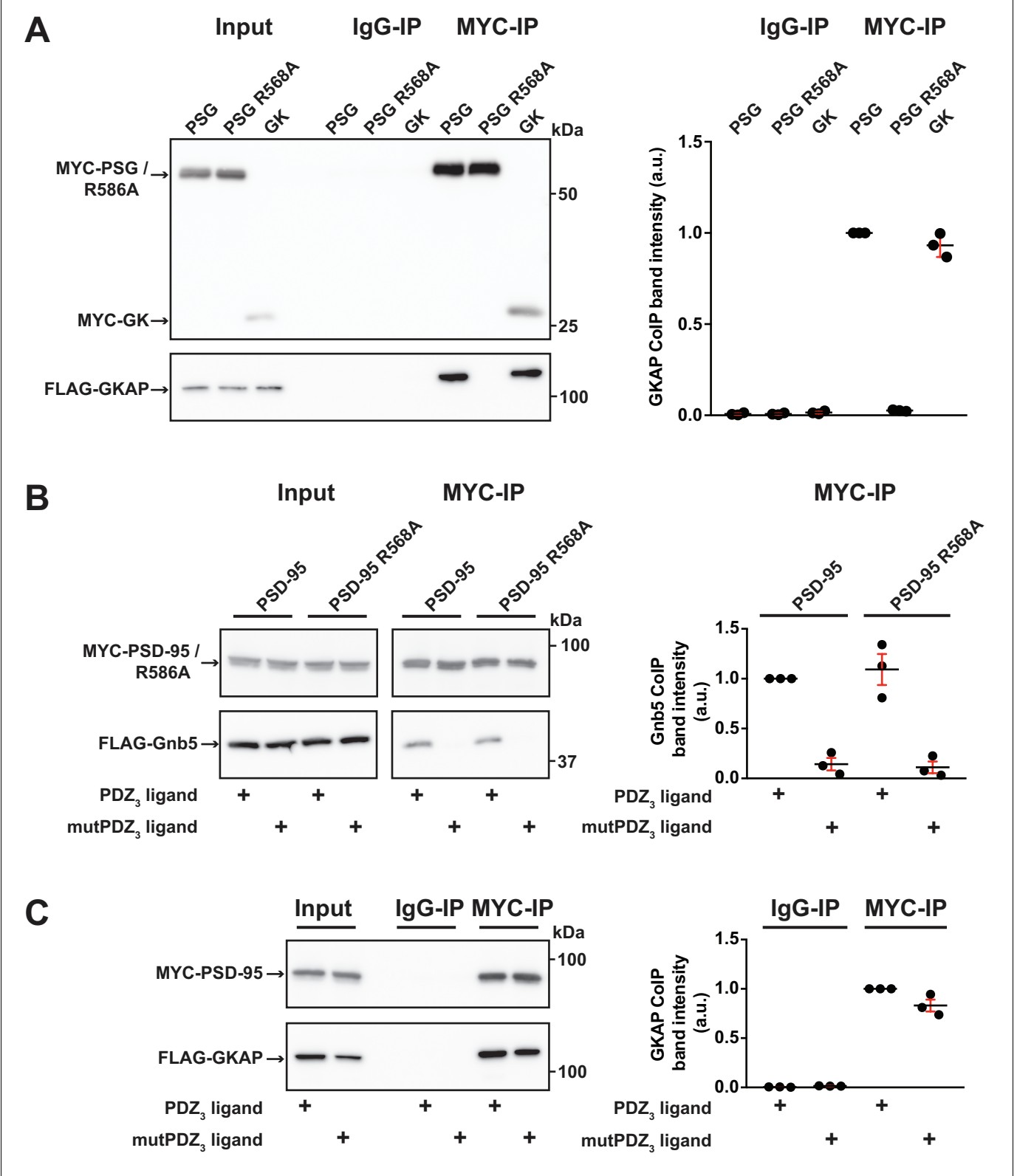

**Figure 5.** GK domain interactions are differentially regulated. For this figure, western blots shown on the left side are representative examples of three independent experiments; the corresponding quantification of coIP band intensities from these three experiments are shown in the dot plots on the right side indicating mean values ± SEM. (**A**) MYC-PSG, MYC-PSG R568A and MYC-GK were coexpressed with FLAG-GKAP. Following MYC IP, precipitated proteins were analysed by western blot. GKAP coprecipitated with PSG and GK domain constructs. The GK domain mutant PSG R568A

*Figure 5 continued on next page*

*Figure 5 continued*

was not able to bind GKAP. (**B**) Following coexpression of MYC-PSD-95 or MYC-PSD-95 R568A with FLAG-Gnb5, together with either CRIPT-derived PDZ$_3$ or mutPDZ$_3$ ligand, proteins were precipitated with αMYC-antibody and analysed by western blot. Gnb5 coIP with either PSD-95 or PSD-95 R568A was efficiently promoted by the presence of PDZ$_3$-binding ligand, irrespective of the GK domain mutation R568A. (**C**) CoIP of MYC-PSD-95 and FLAG-GKAP together with either CRIPT-derived PDZ$_3$ or mutPDZ$_3$ ligand constructs and analysis of precipitated proteins by western blot with antibodies to the corresponding tags. The presence of CRIPT-derived PDZ$_3$ ligands in the lysate had almost negligible effect on PSD-95 GKAP interaction.
DOI: https://doi.org/10.7554/eLife.41299.013

The following source data is available for figure 5:

**Source data 1.** Source data for *Figure 5*.
DOI: https://doi.org/10.7554/eLife.41299.014

interaction during GABA$_B$ – GIRK channel signalling, or if PSD-95 associates with a different pool of Gnb5 that acts in other processes.

Our data indicate that the Gnb5 - PSD-95 interaction is positively regulated by ligand binding to the third PDZ domain of PSD-95. In order to understand how this occurs, it is important to note that in MAGUK scaffold proteins, the SH3 and GK domains interact directly, and together they form a unique structure that sets them apart from SH3 and GK domains found independently in other protein families (*Tavares et al., 2001*). Indeed, PSD-95 SH3 and GK domains, when expressed independently, bind each other efficiently (*McGee and Bredt, 1999*; *Shin et al., 2000*). Likewise, in line with this structural model, mutations that disrupt the interface where these two domains contact each other can have detrimental effects on protein function (*McGee and Bredt, 1999*; *Shin et al., 2000*). Also relevant is the fact that the SH3 domain has been reported to be an allosteric regulator or inhibitor of GK domain binding function, not only by direct contact with the adjacent GK surface but also by regulating the conformation of distant GK domain surfaces (*Marcette et al., 2009*). It is possible that binding of a ligand to the PSD-95 PDZ$_3$ domain influences precisely this function of the neighbouring SH3 domain and thus indirectly regulates GK interactions at distant sites. Alternatively, it is conceivable that regulation via PDZ$_3$ ligand binding results in a conformational change that loosens the natural SH3-GK structure, thereby freeing up the SH3-interacting surface of the GK domain for other protein-protein interactions. Nevertheless, in both possible scenarios, binding of a ligand to the adjacent PDZ$_3$ domain would release the GK domain from its regulation by the interacting SH3 domain. In order to explore these two possibilities, we took advantage of the established GK interactor GKAP and we compared Gnb5 and GKAP with regard to PSD-95 binding. By introducing

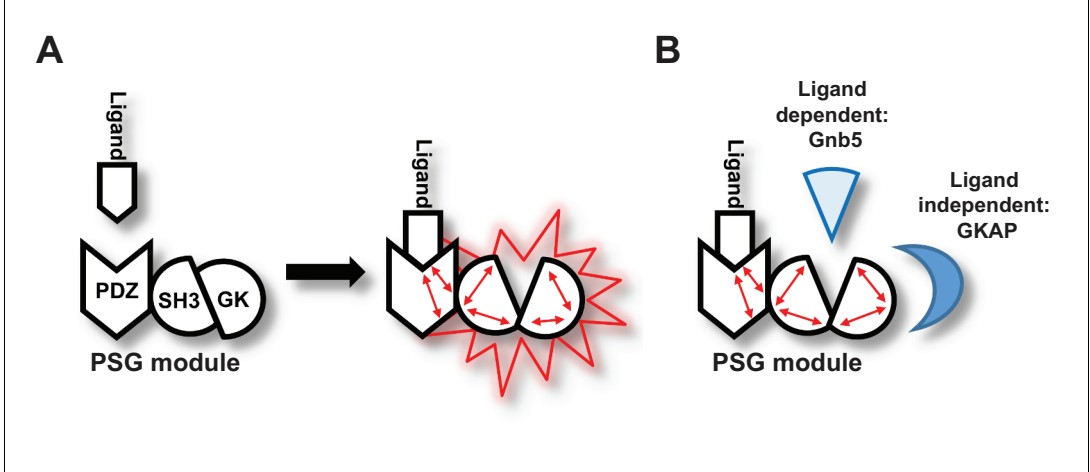

**Figure 6.** Graphical Summary. (**A**) PSD-95 C-terminal domains (PSG module) functionally cooperate and regulate homotypic and heterotypic complex formation. We propose that CRIPT-derived PDZ$_3$ ligand binding to the PDZ$_3$ domain induces a loosening of the intramolecular SH3-GK interaction. This 'open' conformation is then able to initiate subsequent oligomerisation and protein binding. (**B**) Model of CRIPT-derived PDZ$_3$ ligand-dependent and ligand-independent binding to the PSD-95 C-terminal SH3-GK domain tandem. Ligand - PDZ$_3$ domain binding facilitates association with Gnb5.
DOI: https://doi.org/10.7554/eLife.41299.015

a mutation (R568A) in the canonical GMP-binding region of PSD-95, we were able to completely abolish GKAP binding to the PSD-95 GK domain. The GKAP - PSD-95 interaction, however, was not influenced by ligand binding to PDZ$_3$. This result suggests that the canonical GMP-binding region in the GK domain is *not* allosterically regulated by PDZ$_3$ ligand binding. For Gnb5, we observed the opposite pattern: First of all, the R568A mutation had no effect on the Gnb5 - PSD-95 interaction, suggesting that Gnb5 might occupy a different GK domain surface for interaction – one that does not directly overlap with the canonical GMP-binding site responsible for the GKAP interaction. Second, the Gnb5 - PSD-95 association, unlike the GKAP - PSD-95 interaction, is strongly dependent on ligand binding to PDZ$_3$, which provides further support for the idea that Gnb5 binds the PSD-95 GK domain away from the GMP-binding site.

We propose that the Gnb5 - PSD-95 interaction is regulated by a modular allosteric mechanism: the SH3 domain exerts inhibitory activity on the GK domain binding capacity by competing directly with Gnb5 for interaction surfaces. Our data are in line with a model in which the PSD-95 SH3-GK domain tandem undergoes structural rearrangements upon binding of a CRIPT-derived PDZ$_3$ ligand to the adjacent PDZ$_3$ domain, and these changes free up the GK domain for interactions with selected proteins.

Our study highlights three interesting avenues for further study. Exploring how Gnb5 functionally links PSD-95 to GIRK channels, as discussed above, is of primary interest. In addition, the idea that other MAGUKs might be subject to similar ligand-induced alterations and subsequent modulation of protein interactions is relevant. Data from other groups suggest that SynGAP, which also binds to PSD-95 PDZ$_3$, specifically influences conformational coupling between PDZ$_3$ and the SH3-GK module, while the related SAP102 does not undergo the same mechanistic alterations (*Zeng et al., 2018*). It will be interesting to explore if the intramolecular SH3-GK assembly of synaptic MAGUKs is generally regulated by ligand binding to PDZ$_3$, as we observe for PSD-95. It is likewise of interest to investigate in more detail how different PDZ$_3$-binding ligands compare to full-length CRIPT and NLGN1 in how they induce changes within the SH3-GK assembly of PSD-95, and how these changes influence homotypic and heterotypic PSD-95 complex formation. The temporal and spatial expression of PDZ$_3$-binding proteins could potentially steer the formation and composition of specific PSD-95-associated complexes. Moreover, kinase-mediated signalling in response to changes in synaptic activity can also influence binding affinity of ligands for PDZ$_3$ domains (*Walkup et al., 2016*), highlighting another mechanistic level at which PSD-95 complex formation can be regulated by biological processes. These ideas will be explored in future investigations.

## Materials and methods

### DNA constructs

Full-length rat PSD-95 (NM_019621) was cloned into pCMV-Tag3A, to obtain MYC-PSD-95. Arginine 568 was exchanged to Alanine by site-directed mutagenesis to generate MYC-PSD-95 R568A.

PSD-95-YN was generated by a PCR based strategy: amino acids 1–724 of PSD-95 were fused to a flexible 3x(GGGGS) linker followed by amino acids 1–154 of EYFP and an HA-tag. PSD-95-YC was generated accordingly by fusing amino acids 1–724 of PSD-95 to a flexible 3x(GGGGS) linker followed by amino acids 155–238 of EYFP and a MYC-tag. Leucine 460 was exchanged to proline by site-directed mutagenesis to generate PSD-95-YN L460P.

MYC-PSG was generated by cloning a fragment that encodes amino acids 247–724 of PSD-95 into pCMV-Tag3A. MYC-PSG L460P and MYC-PSG R568A were generated by PCR based site-directed mutagenesis. FLAG-SH3-GK and FLAG-GK constructs were generated by cloning fragments that encode amino acids 403–724 (SH3-GK) and amino acids 504–724 (GK) of PSD-95 into pCMV-Tag2A. MYC-GK was generated by cloning a fragment that encodes amino acids 504–724 of PSD-95 into pCMV-Tag3A.

GST-SH3-GK and GST-GK constructs were generated by cloning fragments that encode amino acids 403–724 (SH3-GK) and amino acids 504–724 (GK) of PSD-95 into pGEX-6P-1 (GE Healthcare).

Full-length rat Neuroligin-1 (NLGN1, NM_053868.2) was cloned into pcDNA3.1 and an HA-tag (YPYDVPDYA) was inserted following the signal sequence (between amino acid 45 and 46) by PCR mutagenesis. The C-terminal PDZ ligand motif (TTRV) was mutated to abolish PDZ domain binding by introducing two alanine residues (T**A**R**A**) to generate a mutNLGN1 construct.

Full-length rat SynCAM1 (XM_006242948) was cloned into pCMV-Tag2A and an HA-tag (YPYDVPDYA) was inserted following the signal sequence (between amino acid 47 and 48) by PCR mutagenesis.

CRIPT-derived $PDZ_3$ ligand constructs were generated by fusing an HSV-tag (QPELAPEDPED) to mCherry followed by a flexible 3x(GGGGS) linker and 10 amino acids (DTKNYKQTSV) referring to the PDZ ligand CRIPT. MutPDZ ligand constructs were generated by mutating the C-terminal QTSV motif to Q**AS**A.

Full-length rat Gnb5 (NM_031770) was cloned into pCMV-Tag2A to generate FLAG-Gnb5. A shortGnb5 construct was generated by cloning a fragment that encodes amino acids 34–353 of Gnb5 into pCMV-Tag2A. Gnb5-Clover was generated by cloning Gnb5 into pEYFP-N1. In a subsequent cloning step EYFP was exchanged for Clover. Full-length mouse GKAP (NM_001360665) was cloned into pCMV-Tag2A to generate FLAG-GKAP.

## Cell culture and transfection

COS-7 (DSMZ no. ACC 60) and HEK-293T (DSMZ no. ACC 635) cells were purchased from the Leibniz-Institut DSMZ - Deutsche Sammlung von Mikroorganismen und Zellkulturen GmbH. HEK-293T (DSMZ no. ACC 635) cell identity was confirmed by fluorescent nonaplex PCR of short tandem repeat markers by DSMZ. COS-7 (DSMZ no. ACC 60) cell identity was confirmed by Hinf I-$(gtg)_5$ DNA profiling by DSMZ. Both cell lines were tested negative for mycoplasma. Cultured cells were maintained in DMEM containing 10% FBS, PEN-STREP (1000 U/ml) and 2 mM L-glutamine. Cells were transfected with Lipofectamine 2000 Reagent (Invitrogen) according to the manufacturer's protocol and harvested for subsequent experimental procedures 20–24 hr post transfection.

## Bimolecular fluorescence complementation (BiFC) assay and flow cytometry

HEK-293T cells were cultured in 12 well plates and transfected with the respective expression construct combinations. Prior to analysis by flow cytometry (BD FACS Calibur) the cells were incubated for 60 min at room temperature to promote fluorophore formation. Cells were harvested by gently rocking the culture dishes and washing with PBS/10% FBS. The HEK-293T cell population was identified in the forward vs. side scatter plot and a gate was defined for subsequent analyses. 10,000 single-cell events for each construct combination were measured and fluorescence was quantified (BD CellQuest and FlowJo).

## Coimmunoprecipitation

Transfected COS-7 cells were harvested 20–24 hr post transfection, resuspended in lysis buffer (50 mM Tris-HCl, 100 mM NaCl, 0.1% NP40, pH 7.5/1 ml per T75 flask) and lysed using a 30-gauge syringe needle. Lysates (1 ml) were cleared by centrifugation and incubated with 2 μg of the appropriate antibody (mouse αGFP antibody [Roche], mouse αMYC [Clontech], or normal mouse IgG [Santa Cruz]) for 3 hr followed by a centrifugation at 20,000 x *g*. Supernatants were incubated with 30 μl Protein G-Agarose (Roche) per ml and washed three times with lysis buffer.

## Western blot

Immunocomplexes were collected by centrifugation, boiled in SDS sample buffer, and separated by 10% Tricine-SDS-PAGE (*Schägger, 2006*). Proteins were blotted onto a PVDF membrane (0.2 μm pore size, Bio-Rad) by semidry transfer (SEMI-DRY TRANSFER CELL, Bio-Rad). Membranes were blocked (PBS/0.1% Tween 20/5% dry milk) and incubated overnight with the primary antibody (1:5000). After incubation with the respective horseradish peroxidase (HRP)-conjugated secondary antibody (1:5000), blots were imaged using chemiluminescence HRP substrate (Western Lightning Plus ECL, Perkin Elmer) and a luminescent image analyser (ImageQuant LAS 4000 mini, GE Healthcare). The following primary antibodies were used for protein detection: αFLAG M2-HRP (mouse, A8592, Sigma), αGnb5 (rabbit, ab185206, Abcam), αMYC (rabbit, 2272S, Cell Signalling). Secondary antibodies: αMouse-HRP (115-035-003, Dianova), αRabbit-HRP (111-035-003, Dianova). All western blots shown are representative results from individual pull-down experiments that have been replicated at least three times with similar outcome.

For western blot quantification, coIP band intensities were measured with the ImageQuant TL 1D v8.1 software (GE Healthcare). In each lane, the band detection areas were set to the same size and the background was automatically corrected using the 'rubberband' method. The resulting band volumes (total sum of pixel intensities) were normalised within each experiment to the respective internal control lane and the results represent relative fold changes (mean ±SEM). For a quantitative view of the variations between individual experiments, the resulting values of three experiments were combined and represented as dot plots.

## Isolation of proteins from the crude synaptosome fraction and GST pull-down

One rat brain (Wistar, 2 g) was used to isolate the crude synaptosome fraction with Syn-PER reagent (Thermo Scientific) according to the manufacturer's manual. In brief, one rat brain was homogenised in 20 ml Syn-PER with a Dounce tissue grinder on ice. The resulting homogenate was centrifuged at 4°C for 10 minutes / 1200 x *g*. The supernatant was transferred to a new centrifuge tube and centrifuged at 4°C for 20 minutes / 15,000 x *g*. The resulting crude synaptosome pellet was solubilised in 10 ml PBS/1% Triton X-100 and cleared by centrifugation.

GST-GK and GST-SH3-GK constructs were expressed in *E.coli* BL21 DE3 and purified according to the manufacturer's manual (GST Gene Fusion System, GE Healthcare). 30 µl of glutathione agarose (Pierce) was loaded with GST-GK or GST-SH3-GK proteins and incubated for 3 hr with solubilised proteins. The beads were washed three times with PBS/1% Triton X-100 and further processed for SDS-PAGE.

## Sample preparation and liquid chromatography-mass spectrometry (LC-MS)

Proteins were eluted from the matrix by incubation with SDS sample buffer for 5 min at 95°C and subsequently separated by SDS-PAGE (10% Tricine-SDS-PAGE). Coomassie-stained lanes were cut into 12 slices and in-gel protein digestion and $^{16}O/^{18}O$-labeling was performed as described previously (*Kristiansen et al., 2008*; *Lange et al., 2010*). In brief, corresponding samples were incubated overnight at 37°C with 50 ng trypsin (sequencing grade modified, Promega) in 25 µl of 50 mM ammonium bicarbonate in the presence of heavy water (Campro Scientific GmbH, 97% $^{18}O$) and regular $^{16}O$-water, respectively. To prevent oxygen back-exchange by residual trypsin activity, samples were heated at 95°C for 20 min. After cooling down, 50 µl of 0.5% TFA in acetonitrile was added to decrease the pH of the sample from ~8 to ~2. Afterwards, corresponding heavy- and light-isotope samples were combined and peptides were dried under vacuum. Peptides were reconstituted in 10 µl of 0.05% (v/v) TFA, 2% (v/v) acetonitrile, and 6.5 µl were analysed by a reversed-phase capillary nano liquid chromatography system (Ultimate 3000, Thermo Scientific) connected to an Orbitrap Velos mass spectrometer (Thermo Scientific). Samples were injected and concentrated on a trap column (PepMap100 C18, 3 µm, 100 Å, 75 µm i.d. ×2 cm, Thermo Scientific) equilibrated with 0.05% TFA, 2% acetonitrile in water. After switching the trap column inline, LC separations were performed on a capillary column (Acclaim PepMap100 C18, 2 µm, 100 Å, 75 µm i.d. ×25 cm, Thermo Scientific) at an eluent flow rate of 300 nl/min. Mobile phase A contained 0.1% formic acid in water, and mobile phase B contained 0.1% formic acid in acetonitrile. The column was pre-equilibrated with 3% mobile phase B followed by an increase of 3–50% mobile phase B in 50 min. Mass spectra were acquired in a data-dependent mode utilising a single MS survey scan (m/z 350–1500) with a resolution of 60,000 in the Orbitrap, and MS/MS scans of the 20 most intense precursor ions in the linear trap quadrupole. The dynamic exclusion time was set to 60 s and automatic gain control was set to $1 \times 10^6$ and 5.000 for Orbitrap-MS and LTQ-MS/MS scans, respectively.

### Proteomic data analysis

Identification and quantification of $^{16}O/^{18}O$-labelled samples was performed using the Mascot Distiller Quantitation Toolbox (version 2.6.3.0, Matrix Science). Data were compared to the SwissProt protein database using the taxonomy *rattus* (August 2017 release with 7996 protein sequences). A maximum of two missed cleavages was allowed and the mass tolerance of precursor and sequence ions was set to 10 ppm and 0.35 Da, respectively. Methionine oxidation, acetylation (protein N-terminus), propionamide (C), and C-terminal $^{18}O_1$- and $^{18}O_2$-isotope labeling were used as variable

modifications. A significance threshold of 0.05 was used based on decoy database searches. For quantification at protein level, a minimum of two quantified peptides was set as a threshold. Relative protein ratios were calculated from the intensity-weighted average of all peptide ratios. The median protein ratio of each experiment was used for normalisation of protein ratios. Only proteins that were quantified in all three replicates with a standard deviation of <2 were considered. Mean protein L/H ratios (GST-GK/GST-SH3-GK) from all three replicates were calculated. Known contaminants (e.g. keratins) and the bait protein were removed from the protein output table.

## Live cell microscopy
HEK-293T cells were seeded in 35 mm FluoroDishes (World Precision Instruments) and triple-transfected with PSD-95-YN, PSD-95-YC and Neuroligin-1 expression constructs. Images were acquired using a spinning disk confocal microscope (Nikon CSU-X).

## Immunofluorescence and confocal microscopy
Mixed cultures of primary hippocampal neurons were generated as reported earlier (*Rademacher et al., 2016*). Briefly, E18 Wistar pups were decapitated, and hippocampi were isolated and collected in ice-cold DMEM (Lonza). Single cell solution was generated by partially digestion (5 min at 37°C) with Trypsin/EDTA (Lonza). The reaction was stopped by adding DMEM/10% FBS (Biochrom) following a subsequent washing with DMEM. Tissue was then resuspended in neuron culture medium (Neurobasal supplemented with B27 and 500 µM glutamine) and mechanically dissociated. Neurons were plated at ~$10^5$ cells/cm$^2$ on coverslips coated with poly-D-Lysine and laminin (Sigma). One hour after plating, cell debris was removed and cultures were maintained in a humidified incubator at 37°C with 5% CO$_2$. The hippocampal neurons were fixed at DIV21 with 4% PFA in PBS for 10 min at RT and permeabilised with 0.2% Triton-X in PBS for 5 min. After blocking for 1 hr at RT with blocking solution (4% BSA in PBS) the primary antibodies were incubated overnight at 4°C diluted 1:500 in blocking solution. Secondary antibodies were diluted 1:1000 in blocking solution and incubated for 1 hr at RT. Coverslips were mounted with Fluoromount G and images were acquired with a Leica laser-scanning confocal microscope (Leica TCS-SP5 II, 63x objective). Primary antibodies: αGnb5 (rabbit, ab185206, Abcam), αPSD-95 (mouse, 75–028, NeuroMab), αMAP2 (mouse, 05–346, Millipore), αMAP2 (guinea pig, 188004, Synaptic Systems), αSynapsin (guinea pig, 106004, Synaptic Systems). Secondary antibodies: αRabbit Alexa Fluor 488 (A-21441, Invitrogen), αGuinea pig Alexa Fluor 568 (Thermo Fisher), αGuinea pig Alexa Fluor 405 (ab175678, Abcam), αMouse Alexa Fluor 568 (A-11031, Life Technologies), αMouse Alexa Fluor 405 (A-31553, Invitrogen).

## Laboratory animal handling
All animals used were handled in accordance with the relevant guidelines and regulations. Protocols were approved by the 'Landesamt für Gesundheit und Soziales' (LaGeSo; Regional Office for Health and Social Affairs) in Berlin and animals reported under the permit number T0280/10.

## Acknowledgements
We would like to thank Bettina Schmerl and Hanna Zieger for helpful comments and support, Melanie Fuchs for technical assistance, and the Advanced Medical Bioimaging Core Facility at the Charité Berlin for support with spinning disk microscopy imaging. This work was funded by the DFG (SH 650/2-1, Collaborative Research Centres SFB 958/SFB 665 and Excellence Cluster NeuroCure EXC257).

## Additional information

### Funding

| Funder | Grant reference number | Author |
| --- | --- | --- |
| Deutsche Forschungsgemeinschaft | EXC257 | Stella-Amrei Kunde Sarah A Shoichet |

| Deutsche Forschungsgemeinschaft | SFB958 | Markus C Wahl<br>Christian Freund<br>Sarah A Shoichet |
| --- | --- | --- |
| Deutsche Forschungsgemeinschaft | SH650/2-1 | Sarah A Shoichet |

The funders had no role in study design, data collection and interpretation, or the decision to submit the work for publication.

## Author contributions

Nils Rademacher, Conceptualization, Investigation, Writing—original draft, Writing—review and editing; Benno Kuropka, Stella-Amrei Kunde, Investigation, Writing—review and editing; Markus C Wahl, Christian Freund, Conceptualization, Funding acquisition, Writing—review and editing; Sarah A Shoichet, Conceptualization, Funding acquisition, Writing—original draft, Writing—review and editing

## Author ORCIDs

Nils Rademacher http://orcid.org/0000-0003-0625-6798
Stella-Amrei Kunde http://orcid.org/0000-0003-1493-9097
Sarah A Shoichet http://orcid.org/0000-0003-4933-7846

## Ethics

Animal experimentation: All animals used were handled in accordance with the relevant guidelines and regulations. Protocols were approved by the 'Landesamt für Gesundheit und Soziales' (LaGeSo; Regional Office for Health and Social Affairs) in Berlin and animals reported under the permit number T0280/10.

## Decision letter and Author response

Decision letter https://doi.org/10.7554/eLife.41299.018
Author response https://doi.org/10.7554/eLife.41299.019

## Additional files

### Supplementary files

• Transparent reporting form
DOI: https://doi.org/10.7554/eLife.41299.016

### Data availability

All relevant data generated or analysed during this study are included in the manuscript as source data files.

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
