## [Decision Letter]

Thank you for submitting your article "Intramolecular domain dynamics regulate synaptic MAGUK protein interactions" for consideration by *eLife*. Your article has been reviewed by three peer reviewers, one of whom is a member of our Board of Reviewing Editors, and the evaluation has been overseen by Richard Aldrich as the Senior Editor. The following individual involved in review of your submission has agreed to reveal his identity: Matthieu Sainlos (Reviewer #2).

The reviewers have discussed the reviews with one another and the Reviewing Editor has drafted this decision to help you prepare a revised submission.

Summary:

In this study the authors build on earlier studies by themselves and others showing that binding of ligands to the third PDZ domain of PSD-95 changes the conformation of the SH3-GK domain. Other studies have shown that a portion of the GK domain becomes more exposed upon binding of ligands to PDZ3, and this appears to increase the probability of formation of PSD-95 dimers by "domain-swapping." The new finding in this study is that opening up of the GK domain exposes binding sites for additional proteins to the GK site. The authors carry out a "proteomic" study to identify possible interacting proteins that bind to the GK site alone but not to the tandem SH3-GK domain. They identify several G-protein subunits as potential regulated binders, and follow up with a study of the binding of gnb5, a β subunit that is known to activate GIRK channels downstream of GABAB receptors. They confirm that this interaction is regulated by binding of a PDZ domain ligand and that it does not require a critical arginine residue in the GK binding groove (unlike a characterized GK partner, GKAP). These results indicate that there are at least two different binding modes in place for the GK domain. This study is interesting for two reasons. It identifies a regulated binding site on the GK domain of PSD-95 that is apparently blocked by interaction with the SH3 domain in the monomeric form of PSD-95. The study shows that gnb5 can bind to this site, suggesting that PSD-95 may, under particular conditions, localize gnb5 near synaptically located GABAB receptors at excitatory synapses, possibly increasing the probability of regulation of GIRK channels by nearby release of GABA.

Essential revisions:

There are some major concerns that must be addressed before the study can be published in *eLife*.

1) In most of the figures, results are shown as single immunoblots without quantification, an indication of how often they were reproduced, or the size of the standard error. In general it is not clear how often each individual experiment was repeated. This should clearly be stated for each figure. Quantification and statistical measures over several experiments are necessary for all of these instances in order to ensure the reproducibility and rigor of the findings before they can be published in *eLife*.

2) Figure 1 lacks at least one critical control. The authors need to show and quantify the fluorescent signal When WT/WT split EYFP and WT/L460 split EYFP are expressed without a PDZ binding ligand. In the absence of this control it is not possible to assess with confidence the size of the effect of a PDZ ligand. The mutNLGN1 is an important control, but it is not adequate by itself. In addition, in Figure 1C, mutating the NLGN1 ligand for PDZ3 reduces signal by only ~40%. The mutation of the TTRV sequence to TARA would be expected to mostly if not completely abrogate binding of this ligand to PDZ3. If so then there should be a nearly 100% loss of the signal. The authors need to address to which degree the TARA really reduces binding and, if it is a complete abrogation, explain why the effect in that experiment is only partial. Also, Shin et al., 2000, as cited) find that the L460P mutation does not prevent multimerization of PSD-95 in contrast to what appears to be the case in Figure 1C. The authors need to address this discrepancy. In fact, the authors conclude later that the open conformation (as would be induced by either binding of ligand to PDZ3 or the L460P mutation) is required for binding of Gnb5. Why the L460P mutation does not result in any multimerization of PSD-95 in Figure 1C is unclear.

3) The authors tend to overgeneralize/lack precision on some key elements. Their initial article (Rademacher et al., 2013) only showed that PSD-95 and not synaptic MAGUK proteins oligomerized upon PDZ domain ligand binding. Other reports indicate different behavior within the PSD-95 family with respect to oligomerization (Zeng et al., 2018 – and not 2017). Similarly, the authors often use the term PDZ3 ligand leading the reader to believe that any PDZ3 ligand will have the same effect. However they have only used two ligands -CRIPT and Neuroligin- and again other reports indicate a difference between CRIPT and another PDZ3 ligand, SynGAP, on their effect on the PSG module (Zeng et al., 2018). Of note, the molecular mechanism by which SynGAP interaction leads to PSD-95 oligomerization is clearly presented in that same study, despite what is written in the discussion (referring to an older citation).

4) Along these lines, considering that CRIPT and SynGAP led to different results for PSD-95 oligomerization in Zeng et al., 2018, it would be interesting to clarify how the present results compare to the one obtained with the SynGAP PDZ3 ligand. Another point is that while CRIPT is, to my knowledge, a monomer, full-length Neuroligin is a dimer which complicates the interpretation of the results of the first section and raises the question if the same oligomerization mechanism is observed.

5) The claim that Gnb5 and GKAP occupy different subdomains is not supported by totally convincing data. The results clearly show a different mode of interaction but do not exclude the fact that it could be the same subdomain. Simultaneous co-precipitation of the two GK partners, if successful, could support such a claim.

6) The method that the authors use to isolate "synaptic proteins" using a reagent sold by Thermo-Fisher has not been verified in any peer-reviewed publication. A search of the online references listed on the Thermo-Fisher website does not contain any method that can reliably enrich for synaptic proteins without the use of a density gradient. The inadequacy of the method is revealed, for example, by the high level of ribosomal proteins listed in the author's Figure 2—source data 1. In subsection “The SH3-GK assembly state influences PSD-95 interactions with synaptic proteins”, the authors cite the overlap between their list of proteins and that of Wilkinson et al., 2017. However, Wilkinson et al. did not use a method that enriches for synaptic proteins. They simply extracted a crude membrane pellet from a brain homogenate with detergent to obtain detergent insoluble proteins. The extracted pellet would contain microsomes and mitochondria, in addition to synaptosomes. This fraction is not a postsynaptic density fraction because it skips the step of isolating synaptosomes from the brain homogenate. For the continued integrity of the literature, the authors should eliminate statements that indicate that they are working with "synaptic proteins" or "postsynaptic density" proteins. They should eliminate the reference to the Wilkinson paper completely as it does not support their statement that they are working with synaptic proteins. The authors have wisely chosen to work with a protein that has been shown to be located in the excitatory postsynaptic compartment, and to play an important role there (regulation of GIRK channels by GABAB receptors). They have verified the synaptic location of the particular protein they have studied. Nonetheless, all claims to have started with anything other than a detergent insoluble extract of a crude brain membrane fraction should be removed.

7) There is not enough information in the Materials and methods section about how flow cytometry was performed and no actual plots of data distribution are shown as would be customary for such analysis, including the presumed existence of various populations of cells. Did the authors stain for cell viability (like propidium iodide) to exclude defective cells?

[Editors' note: further revisions were requested prior to acceptance, as described below.]

Thank you for resubmitting your work entitled "Intramolecular domain dynamics regulate synaptic MAGUK protein interactions" for further consideration at *eLife*. Your revised article has been favourably evaluated by Richard Aldrich (Senior Editor), and three reviewers, one of whom is a member of our Board of Reviewing Editors.

The manuscript has been improved but there are some remaining issues that need to be addressed before acceptance, as outlined below:

In this first revision, the authors have responded adequately to some, but not all of the major comments on the original manuscript. They have included quantitation and statistical analysis of all of their experiments as appropriate. They have included a new control (SynCAM1) in their transfections to strengthen the conclusion that the interaction of the two halves of EYFP fused to neuroligin and transfected into 293 cells requires interaction of neuroligin with a PDZ domain of PSD-95.

However, the authors have not responded adequately to the criticism that they have over-generalized and lack precision in their writing. There are two aspects of this problem.

1) The authors conflate the finding that the CRIPT 10-mer "opens-up" the GK domain exposing a new domain that binds to Gnb5, with the generalization that the CRIPT 10-mer promotes multimerization of PSD-95 (which they believe also leads to dimerization of tagged neuroligin). The implication is that the multimerization of PSD-95 arises from the "opening up" of the GK and SH3 domains so that they can interact in trans with other proteins. This generalization conflicts with the finding reported by Zeng et al. in Journal of Molecular Biology (2018) that binding of a peptide containing the final 17 residues of CRIPT (CRIPT 17-mer) to PDZ3 does NOT lead to multimerization of PSD-95; whereas binding of a peptide containing the final 15 residues of synGAP DOES result in multimerization of PSD-95.

Communication about this issue may have been obfuscated by the fact that the authors have mis-stated the year of this Zeng et al. publication as 2017. The authors need to fix the citation of the J. Mol. Bio. paper by Zeng et al. by changing the year to 2018. Note that a later Cell paper published by Zeng et al. in 2018 on a similar topic is actually not relevant to the present study and should not be cited.

In the Zeng et al. 2018 J. Mol. Biol. paper, Zeng et al. used size-exclusion chromatography and NMR to show that binding of the synGAP 15-mer to PDZ3 (including a short downstream α helix) causes dimerization of the PSD-95 construct; whereas, binding of the CRIPT 17-mer does not cause dimerization of PSD-95. The reviewers can think of two possible reasons for the discrepancy between their results and the present manuscript. One is that Zeng et al. used a CRIPT 17-mer; whereas the present authors used a CRIPT 10-mer. Perhaps the two peptides produce different results with respect to "opening-up" of the GK domain. If this is the case, it calls into question the in vivo significance of the finding with the CRIPT 10-mer. The other possible explanation is that the binding of either of the two CRIPT peptides to PDZ3 of PSD-95 causes a change in conformation of the SH3-GK domains such that a new binding site for Gnb5 and other proteins is made available; however, the conformational change does not permit full dimerization of PSD-95 through the domain-swapping trans association. In other words, CRIPT peptides may produce a different change in the structure of the SH3-GK domain than does the synGAP 15-mer. In the present manuscript, the authors have not used any methodology that would directly detect dimerization of the PSD-95 construct by the domain-swapping trans interaction. They have assumed that fluorescence resulting from interaction of the two halves of EYFP linked to neuroligin results from oligomerization of PSD-95. For example, they state:

"In this study, we use a bimolecular fluorescence complementation (BiFC) assay to show that PSD-95 oligomerisation is triggered by PDZ3 ligands and dependent on the C-terminal SH3-GK domain tandem."

To resolve the apparent conflict between their data and that of Zeng et al., the authors should compare the effect of a CRIPT 17-mer in their assays to the effect of the CRIPT 10-mer and test directly for dimerization of the PSD-95 construct after binding of the peptides by size-exclusion chromatography or another method that would unambiguously detect dimerization of PSD-95.

2) The authors continue to inappropriately generalize their results to all PDZ3 binding ligands. For example:

In the Abstract, "Here we show that binding of monomeric PDZ3 ligands to the third PDZ domain of PSD-95 induces functional changes in the intramolecular SH3-GK domain assembly that influence subsequent homotypic and heterotypic complex formation."

"…this interaction is triggered by PDZ3 ligands binding to the third PDZ domain of PSD-95,…"

Introduction section: We have previously shown that synaptic MAGUK proteins oligomerise upon binding of monomeric PDZ3 ligands (ligands that specifically bind to the third PDZ domain) (Rademacher et al., 2013) and speculated that ligand – PDZ3 domain binding induces conformational changes in the C-terminal domains that lead to complex formation."

This generalization to all PDZ3 ligands is repeated in several places in the manuscript. The authors should replace the term PDZ3 ligand with the name of the specific ligand that the authors use in each experiment.

---

## [Author Response]

Essential revisions:There are some major concerns that must be addressed before the study can be published in eLife.1) In most of the figures, results are shown as single immunoblots without quantification, an indication of how often they were reproduced, or the size of the standard error. In general it is not clear how often each individual experiment was repeated. This should clearly be stated for each figure. Quantification and statistical measures over several experiments are necessary for all of these instances in order to ensure the reproducibility and rigor of the findings before they can be published in eLife.

All immunoblots in the Results section of the revised manuscript (Figures 1E, 3A, 3B, 4A, 4B, 4C, 5A, 5B, and 5C) show the results of representative experiments that were performed with identical conditions three times. As recommended by the reviewers, we quantified the 3 experiments and calculated mean intensities plus standard error and included these results as dot plots within the main figures. We have modified the figure legends accordingly and described the quantification procedure in detail in the Materials and methods section. The measured band intensities (original data) are now included as a supplemental table.

2) Figure 1 lacks at least one critical control. The authors need to show and quantify the fluorescent signal When WT/WT split EYFP and WT/L460 split EYFP are expressed without a PDZ binding ligand. In the absence of this control it is not possible to assess with confidence the size of the effect of a PDZ ligand. The mutNLGN1 is an important control, but it is not adequate by itself. In addition, in Figure 1C, mutating the NLGN1 ligand for PDZ3 reduces signal by only ~40%. The mutation of the TTRV sequence to TARA would be expected to mostly if not completely abrogate binding of this ligand to PDZ3. If so then there should be a nearly 100% loss of the signal. The authors need to address to which degree the TARA really reduces binding and, if it is a complete abrogation, explain why the effect in that experiment is only partial.

Indeed these are relevant points and we have addressed them in the revised manuscript. Comparing refolding of splitEYFP halves in the presence and absence of NLGN1 (which involves comparison of a triple transfection with a double transfection) is problematic; PSD-95-splitEYFP expression levels change when there is no third molecule present, thereby making results difficult to compare with those following double transfection. We have therefore designed a more suitable control, in which we substitute NLGN1 with a comparable amount of SynCAM1 DNA in the transfection. SynCAM1, like NLGN1, is a synaptic cell adhesion molecule, but unlike NLGN1, is not able to bind to PSD-95 (Biederer et al., 2002). Instead, it binds to other MAGUKs, including e.g. CASK and MPP2, which harbour class II ligand-binding PDZ domains rather than class I ligand-binding PDZ domains. In an additional set of experiments (done in triplicate and repeated 4 times, as all of our FACS analyses are performed), we have included this control (see new Figure 1C). As expected, SynCAM1 is unable to induce PSD-95 complex formation and subsequent EYFP refolding.

Regarding our mutant NLGN1 control: It is indeed likely that we do not have 100% abrogation of binding to PSD-95 with the TARA mutant, which would explain why we still see significant EYFP refolding of PSD-95-splitEYFP halves in the presence of the NLGN1 molecule with mutations in the PDZ binding motif. This idea is supported by our new Figure 1C, in which we compare the effects of wild-type NLGN1 with that of the synaptic cell adhesion molecule SynCAM1 (described above). In particular for PDZ ligands that have substantial cytosolic sequences (as is the case for NLGN1), it is likely that residues outside of the C-terminal PDZ binding motif influence ligand-PDZ domain binding and/or stability. For several ligand-PDZ domain interactions, including e.g. SynGAP/PSD-95 (Walkup et al., 2016) and Crumbs/PALS1 (Li et al., 2014), this is clearly the case; we expect that related phenomena also apply in the case of NLGN1.

Also, Shin et al., 2000, as cited, find that the L460P mutation does not prevent multimerization of PSD-95 in contrast to what appears to be the case in Figure 1C. The authors need to address this discrepancy. In fact, the authors conclude later that the open conformation (as would be induced by either binding of ligand to PDZ3 or the L460P mutation) is required for binding of Gnb5. Why the L460P mutation does not result in any multimerization of PSD-95 in Figure 1C is unclear.

Shin et al. (Shin et al., 2000) showed that PSD-95 L460P mutant proteins associate with membranes and interact with the potassium channel Kv1.4 (via PDZ domain interactions) indistinguishable from wild-type PSD-95. These results resemble our own observations regarding the subcellular localisation and PDZ-binding properties of PSD-95 L460P (see e.g.(Rademacher et al., 2013)). However, the L460P mutant proteins fail to cluster the Kv1.4 channels in their clustering assay, an interesting observation that is in line with our FACS results in Figure 1, where this single amino acid exchange in the C-terminal SH3 domain also interferes with higher order protein complex formation. In their study, Shin et al. additionally use a coimmunoprecipitation assay to explore PSD-95 multimerisation. In this part of their study, however, they do not investigate the behaviour of full-length PSD-95, which explains why their resulting conclusions differ from ours. Shin et al. demonstrate that the PSD-95 L460P mutant associates efficiently with a short N-terminal PSD-95 variant containing what they refer to as “the N-terminal multimerization domain” (N-PSD-GFP, which consists exclusively of the N-terminal 64 amino acids of PSD-95 fused to GFP); this experiment is not comparable to our FACS assay with full-length PSD-95 variants.

It is perhaps important to point out here that since the study by Shin et al. was published in 2000, it has become clear that this N-terminal region is not the main element involved in PSD-95 multimerisation. Shortly afterwards, McGee et al., 2001, highlighted a different multimerisation mechanism that relies on the C-terminal SH3-GK module (McGee et al., 2001), and our own work (e.g. (Rademacher et al., 2013)), as well as that of other groups (Zeng et al., 2016; Zeng et al., 2017) likewise focusses on multimerisation mechanisms that involve the SH3-GK module instead of the N-terminus of PSD-95. As we point out in the manuscript, the L460P mutation abolishes the intramolecular reaction between the SH3 and GK domains (McGee and Bredt, 1999), and thereby prevents the SH3 domain from interacting with the GK domain in both intra- and intermolecular PSD-95 interactions, thus negatively influencing well-ordered formation of higher order PSD-95 complexes. At the same time, however, such changes free the GK domain from inhibitory effects of the intramolecular SH3 interaction, and thereby facilitate interaction with other proteins, such as Gnb5.

3) The authors tend to overgeneralize/lack precision on some key elements. Their initial article (Rademacher et al., 2013) only showed that PSD-95 and not synaptic MAGUK proteins oligomerized upon PDZ domain ligand binding. Other reports indicate different behavior within the PSD-95 family with respect to oligomerization (Zeng et al., 2018 – and not 2017). Similarly, the authors often use the term PDZ3 ligand leading the reader to believe that any PDZ3 ligand will have the same effect. However they have only used two ligands -CRIPT and Neuroligin- and again other reports indicate a difference between CRIPT and another PDZ3 ligand, SynGAP, on their effect on the PSG module (Zeng et al., 2018).

It is true that this study is focused on PSD-95 and not on synaptic MAGUK proteins in general, and we have made an effort to be more specific with our word choice throughout the revised manuscript. For example, we have now specified that we mean the “MAGUK protein PSD-95” rather than referring simply to “MAGUK protein” at relevant points within the text (see paragraph one of the Introduction).

In order to focus on the effects of ligand binding to PDZ3 rather than general ligand binding to PDZ domains in PSD-95, we have selected ligands that are historically known for interacting predominantly with PDZ3, which is the case for both NLGN1 and CRIPT (Irie et al., 1997; Niethammer et al., 1998). Our cell-based experiments described in Figure 1 build on our previous work on how ligand binding to PDZ domains influences PSD-95 multimerisation (Rademacher et al., 2013). For the cell-based experiments in Figure 1 of this manuscript, we elected to use full-length proteins that have a solid history as postsynaptic interaction partners and therefore selected NLGN1 as our PDZ3 ligand.

In subsequent experiments we took advantage of the CRIPT C-terminus to produce a generic binder of PDZ3, for the purpose of investigating in general how ligand binding at PDZ3 of PSD-95 influences the scaffolding behaviour of this molecule and specifically how it influences the behavior of PSD-95 in intra-and inter-molecular protein-protein interactions. We used the CRIPT C-terminus for this purpose in part because it has a long history as a PSD-95 PDZ3-binding protein, and also because it is the most efficient and specific PDZ3-binding ligand in our hands. In order to make things clearer and prevent misunderstandings, we have now consistently specified throughout the text which PDZ3 ligand we have used for which experiments. We have also added an introductory sentence in the results (Results section entitled “Ligand binding to PSD-95 PDZ3 facilitates an “open” SH3-GK state that frees both domains for binding in trans”) explaining our choice of the CRIPT C-terminus for pursuing the main questions addressed in this study.

Of note, the molecular mechanism by which SynGAP interaction leads to PSD-95 oligomerization is clearly presented in that same study, despite what is written in the discussion (referring to an older citation).

In response to this point, we have modified our discussion to include more detail on the results of Zeng et al., 2017 (see paragraph two of the Discussion and also the final paragraph of the Discussion).

4) Along these lines, considering that CRIPT and SynGAP led to different results for PSD-95 oligomerization in Zeng et al., 2018, it would be interesting to clarify how the present results compare to the one obtained with the SynGAP PDZ3 ligand. Another point is that while CRIPT is, to my knowledge, a monomer, full-length Neuroligin is a dimer which complicates the interpretation of the results of the first section and raises the question if the same oligomerization mechanism is observed.

We acknowledge that NLGN1 is capable of forming dimers. Our cell-based splitEYFP assay includes the scaffold-deficient L460P control that hinders complex formation despite the possible presence of NLGN1 dimers (see Figure 1D), which supports the idea that the observed EYFP refolding reflects complex formation that depends on the functionally intact SH3-GK domain rather than on the dimerisation properties of NLGN1 molecules. This initial result is in line with our previous work with the CRIPT PDZ ligand (Rademacher et al., 2013), but the effects of SynGAP have not been tested yet. A rigorous comparison of the similarities and differences between NLGN, CRIPT, and SynGAP with regard to how they influence the scaffolding properties of PSD-95 in our assay is of interest and certainly worth pursuing further experimentally. In this context, further investigations into how different PDZ binders influence PSD-95 interactions and thereby steer the formation of specific complexes (both temporally and spatially) is also relevant, and we have highlighted these interesting avenues for further study in our modified discussion. We feel, however, that these questions extend beyond the aims and scope of this study, given that our focus here is on ligand-mediated conformational alterations to the PSD-95 molecule and resulting physical consequences, rather than on the complex biology of the precursor steps.

5) The claim that Gnb5 and GKAP occupy different subdomains is not supported by totally convincing data. The results clearly show a different mode of interaction but do not exclude the fact that it could be the same subdomain. Simultaneous co-precipitation of the two GK partners, if successful, could support such a claim.

Indeed our data do not enable us to conclude which precise surface of the GK domain is involved in the interaction with Gnb5. This is a relevant point and one that we are investigating. We are currently using a targeted approach in which we mutate selected amino acids throughout the GK domain sequence with the aim of interfering with the Gnb5-GK interaction. Based on existing structural models of the GK domain, we have already tested several single amino acid changes; so far, however, we have not observed dramatic effects on Gnb5 binding (as we observe for the R568A mutation with regard to GKAP binding). An additional aspect of this approach, which we will also pursue in the future, involves crystallisation of the GK domain in complex with Gnb5 in order to precisely identify the best residues for targeted mutation.

Simultaneous coprecipitation is another method that can be used to explore this idea. We know that after pulldown of the isolated GK domain we can coprecipitate GKAP and Gnb5. Following triple transfection (in which both GKAP and Gnb5 are simultaneously expressed) and pulldown of the GK domain, we expect to find both of these proteins in the coprecipitate, even if they bind to the same surface of GK, as we will pull down sufficient GK molecules for participation in both interactions. In order to prove that the proteins do not compete for the same binding surface, our approach involved a triple transfection of the three constructs and pulldown of Gnb5, followed by analysis of GKAP (or vice versa), i.e. we attempted to isolate the three proteins in a single complex. Although we were indeed able to coprecipitate the GK domain using this strategy, we did not observe formation of a complex in which all proteins could be detected. There are several possible explanations for this. Of course it is possible that Gnb5 and GKAP indeed occupy partially overlapping regions of the GK domain. It is also possible that we simply do not have adequate coprecipitation of the GK domain to enable enrichment of the third binding partner. Pulldown experiments in which we immunoprecipitate the GK domain directly consistently yield substantial amounts of GK in the precipitate; with our triple transfection approach, however, GK levels in the precipitate are low. It is also possible that the presence of a large protein bound to GK at one surface simply precludes binding at a neighbouring site due to stearic hindrance. Indeed, given what is known about these two proteins, we would not necessarily expect them to bind simultaneously to the same PSD-95 molecule in neurons. While GKAP binds to PSD-95 molecules at the core of the PSD, it is plausible that Gnb5 instead links less central PSD-95 molecules with membrane proteins at the periphery of the PSD. These are questions that we intend to pursue in future studies; we have expanded on these ideas in the revised discussion.

Importantly, in response to this issue, we have modified the text slightly to avoid overstatements in the revised manuscript. Specifically, we have changed the title of the final Results subsection from “PSD-95 interactors occupy different GK subdomains” to “GK-domain interactions are differentially regulated”. We have made the same adjustment to the title of Figure 5.

6) The method that the authors use to isolate "synaptic proteins" using a reagent sold by Thermo-Fisher has not been verified in any peer-reviewed publication. A search of the online references listed on the Thermo-Fisher website does not contain any method that can reliably enrich for synaptic proteins without the use of a density gradient. The inadequacy of the method is revealed, for example, by the high level of ribosomal proteins listed in the author's Figure 2—source data 1. In subsection “The SH3-GK assembly state influences PSD-95 interactions with synaptic proteins”, the authors cite the overlap between their list of proteins and that of Wilkinson et al., 2017. However, Wilkinson et al. did not use a method that enriches for synaptic proteins. They simply extracted a crude membrane pellet from a brain homogenate with detergent to obtain detergent insoluble proteins. The extracted pellet would contain microsomes and mitochondria, in addition to synaptosomes. This fraction is not a postsynaptic density fraction because it skips the step of isolating synaptosomes from the brain homogenate. For the continued integrity of the literature, the authors should eliminate statements that indicate that they are working with "synaptic proteins" or "postsynaptic density" proteins. They should eliminate the reference to the Wilkinson paper completely as it does not support their statement that they are working with synaptic proteins. The authors have wisely chosen to work with a protein that has been shown to be located in the excitatory postsynaptic compartment, and to play an important role there (regulation of GIRK channels by GABAB receptors). They have verified the synaptic location of the particular protein they have studied. Nonetheless, all claims to have started with anything other than a detergent insoluble extract of a crude brain membrane fraction should be removed.

Point well taken. We acknowledge that the strategy we have used to generate lysates that contain synaptic proteins is not by any means one that yields pure synaptosomes, and the description of the method we used for protein isolation in our original manuscript was not adequate. Given the reagents and methods we have used, and in light of the reviewers’ concerns, we have reconsidered how we describe our work and how we compare it to data generated by other groups. Importantly, we now describe in more detail in the Materials and methods section the steps involved in generating our protein lysates. We acknowledge that our protocol yields a membrane fraction from brain that is enriched for synaptosomal proteins, but also contains mitochondrial proteins and other non-synaptic membrane contaminants. There is some flexibility across the literature with regard to how people refer to such a preparation. We have opted to maintain our use of the term “crude synaptosome preparation”, and we have corrected the text in the two places where we made the error of referring to “synaptic protein lysates” or “synaptosomes” in the original submission. We additionally deleted “synaptic proteins” from the title of the third subsection of the Results and adjusted the corresponding text of Figure 2A to “crude synaptosome preparation”.

We also acknowledge that the classical cell fractionation method, involving ultracentrifugation and use of a sucrose gradient, is the reliable method for producing a pure synaptosome fraction and subsequent PSD fraction (Carlin et al., 1980). For this reason, we have decided to remove the comparison of our identified proteins with the list generated by Wilkinson et al. Instead we have now compared our list to the list of PSD proteins generated by Baye et al., (Bayes et al., 2012), who used a classical approach for isolating PSD proteins based on the methods described by P. Siekevitz and colleagues (see the third subsection of the Results entitled “The SH3-GK assembly state influences PSD-95 interactions”).

7) There is not enough information in the Materials and methods section about how flow cytometry was performed and no actual plots of data distribution are shown as would be customary for such analysis, including the presumed existence of various populations of cells. Did the authors stain for cell viability (like propidium iodide) to exclude defective cells?

We have now included exemplary FACS plots for every construct combination we have analysed in the manuscript (see results in Figure 1C and Figure 1D) as a Supplementary file (Figure 1—figure supplement 1 FACS). During establishment of the FACS assay, we regularly included a propidium iodide control to identify cells with damaged membranes. Given that we could efficiently exclude these π positive objects in further analyses by taking advantage of gating strategies to define living cells in the forward vs. side scatter plot, we no longer include the propidium iodide control regularly in our experimental plan. Indeed the description of our cell sorting assay was brief in our original submission and much of this information was lacking. In addition to the exemplary FACS plots now attached to Figure 1 (Figure 1—figure supplement 1), our revised manuscript also includes a thorough description of the FACS procedure that includes the details of our gating strategy (see revised Materials and methods).

[Editors' note: further revisions were requested prior to acceptance, as described below.]

The manuscript has been improved but there are some remaining issues that need to be addressed before acceptance, as outlined below:In this first revision, the authors have responded adequately to some, but not all of the major comments on the original manuscript. They have included quantitation and statistical analysis of all of their experiments as appropriate. They have included a new control (SynCAM1) in their transfections to strengthen the conclusion that the interaction of the two halves of EYFP fused to neuroligin and transfected into 293 cells requires interaction of neuroligin with a PDZ domain of PSD-95.However, the authors have not responded adequately to the criticism that they have over-generalized and lack precision in their writing. There are two aspects of this problem.1) The authors conflate the finding that the CRIPT 10-mer "opens-up" the GK domain exposing a new domain that binds to Gnb5, with the generalization that the CRIPT 10-mer promotes multimerization of PSD-95 (which they believe also leads to dimerization of tagged neuroligin). The implication is that the multimerization of PSD-95 arises from the "opening up" of the GK and SH3 domains so that they can interact in trans with other proteins. This generalization conflicts with the finding reported by Zeng et al. in Journal of Molecular Biology (2018) that binding of a peptide containing the final 17 residues of CRIPT (CRIPT 17-mer) to PDZ3 does NOT lead to multimerization of PSD-95; whereas binding of a peptide containing the final 15 residues of synGAP DOES result in multimerization of PSD-95.Communication about this issue may have been obfuscated by the fact that the authors have mis-stated the year of this Zeng et al. publication as 2017. The authors need to fix the citation of the J. Mol. Bio. paper by Zeng et al. by changing the year to 2018. Note that a later Cell paper published by Zeng et al. in 2018 on a similar topic is actually not relevant to the present study and should not be cited.[…]To resolve the apparent conflict between their data and that of Zeng et al., the authors should compare the effect of a CRIPT 17-mer in their assays to the effect of the CRIPT 10-mer and test directly for dimerization of the PSD-95 construct after binding of the peptides by size-exclusion chromatography or another method that would unambiguously detect dimerization of PSD-95.

Our response to point 1 is multifaceted and includes, for clarification, a summary of our current approach and how it differs from our work in previous studies.

A minor aspect of our study focusses on the idea that binding of ligands to the third PDZ domain of PSD-95 can influence PSD-95 multimerisation. This question was the focus of our previously published work (Rademacher et al., 2013), in which we took advantage of the CRIPT PDZ ligand and demonstrated that binding of monomeric CRIPT-derived C-termini to PDZ3 promoted interactions between PSD-95 molecules. In this previous study, we also demonstrated that this multimer formation brought CRIPT-derived PDZ ligand C-termini into close proximity of each other (as shown by refolding of EYFP when CRIPT ligand sequences fused to non-fluorescing EYFP halves were expressed together with PSD-95 or PSD-95 fragments). In the current study, we have used a new BiFC strategy that integrates full-length PSD-95 molecules fused to non-fluorescent EYFP halves – and we have shown that expression of NLGN also promotes PSD-95 multimolecular complex formation. Importantly, we show in this study that this process relies on both successful ligand binding to the PDZ domains AND the integrity of the SH3-GK domain module (see Figure 1). However, the mechanistic details of PSD-95 multimerisation, which in our hands can be induced by both NLGN and CRIPT, is not the focus of our current study.

We observe, in both previous and current studies, that tagged CRIPT-derived C-termini induce changes in the behaviour of the PSD-95 PSG module, and here we have successfully taken advantage of this phenomenon to isolate new GK binders that are regulated by ligand-PDZ domain interactions. It is this idea that is the central theme of this study. We focus specifically on the regulated interaction between PSD-95 and Gnb5, a novel GK domain interactor whose affinity for PSD-95 is regulated by binding of CRIPT-derived ligands to PSD-95 PDZ3. Our CoIP data are in line with the model that the PSD-95 SH3-GK domain module indeed “opens up” in response to binding of CRIPT-derived ligands to PDZ3, and that this conformational change is what enables binding to Gnb5. While it seems plausible that similar mechanisms underly PSD-95 multimer formation, we have not investigated this question directly in this study and therefore have made efforts to remove any misleading generalisations that suggest something different. For example, we have removed the following sentence in the Discussion: “Here, we show that PSD-95 oligomerisation can be induced by binding of monomeric PDZ3 ligands, which then leads to conformational changes within the adjacent C-terminal SH3-GK domain structure.”

Instead we refer directly to our previous work: “In previous work, we showed that binding of CRIPT-derived ligands to PDZ3 of PSD-95 promoted formation of PSD-95 multimers.” Later we summarise our current work: “Here we identify synaptic interactors whose association with PSD-95 is influenced by the conformational state of the PSD-95 C-terminus.”

We have also removed the following general sentence later on in the Discussion: “Via this mechanism, ligand-PDZ3 domain interactions facilitate formation of both homotypic and heterotypic complexes guided by the PSD-95 C-terminal PSG module.”

Finally, we recognize that in in vitro biochemical assays, tagged protein fragments (as we use here) or peptides (as used in Zeng et al., 2018) often replace full-length proteins for simplicity in experimental design and that these strategies are inevitably limited in terms of applicability to the in vivo situation. In order to confirm the in vivo validity of our experiments, and understand why CRIPT peptides do not induce multimerisation of PSD-95 PSG molecules in other labs (Zeng et al. 2018), multiple strategies, that ideally make use of full-length proteins, are required. Certainly, a comparison of how wild-type CRIPT, NLGN, and SynGAP influence PSD-95 complex formation and specifically the interaction between PSD-95 and novel binding partner Gnb5 is of interest, especially in light of the apparent discrepancy between our results and those of Zeng et al. 2018 with regard to how CRIPT-derived fragments influence PSD-95 multimer formation. However, this should be pursued in future studies using wild-type and/or endogenous proteins, rather than using peptides (as done by Zeng et al. 2018) or proteins fragments (as we have done here). These experiments are planned, but they extend beyond the scope of this study.

Finally, we appreciate your clarification regarding the Zeng et al., 2018 references and we have now removed the Zeng et al., 2018 (Cell) references, which is less directly related our study.

2) The authors continue to inappropriately generalize their results to all PDZ3 binding ligands. For example:In the Abstract, "Here we show that binding of monomeric PDZ3 ligands to the third PDZ domain of PSD-95 induces functional changes in the intramolecular SH3-GK domain assembly that influence subsequent homotypic and heterotypic complex formation.""…this interaction is triggered by PDZ3 ligands binding to the third PDZ domain of PSD-95,…"Introduction section: We have previously shown that synaptic MAGUK proteins oligomerise upon binding of monomeric PDZ3 ligands (ligands that specifically bind to the third PDZ domain) (Rademacher et al., 2013) and speculated that ligand – PDZ3 domain binding induces conformational changes in the C-terminal domains that lead to complex formation."This generalization to all PDZ3 ligands is repeated in several places in the manuscript. The authors should replace the term PDZ3 ligand with the name of the specific ligand that the authors use in each experiment.

We acknowledge that we have not been adequately specific in the description of our results with regard to the PDZ ligands used in our experiments, and we have made relevant changes throughout the manuscript in order to indicate very clearly how each experiment was done and which ligand was used. For examples, see:

Figure 1 E legend

Figure 4 title

Figure 4 A and B legends

Figure 5 B and C legends

Figure 6 legend

We have also adjusted statements in which our wording implied that we were referring to PDZ3 ligands in general. For example, in the Abstract, Introduction, Results subsection “Neuroligin-1 binding to PSD-95 PDZ3 domains facilitates oligomerisation guided by the PSG module” and “Binding of a CRIPT-derived PDZ ligand to PSD-95 PDZ3 facilitates an “open” SH3-GK state that frees both domains for binding in trans”, and in numerous other points in the text we have now specified “CRIPT-derived PDZ3 ligands’ instead of using the more general ‘PDZ3 ligands”.